# HnRNPK maintains single strand RNA through controlling double-strand RNA in mammalian cells

Sagar Mahale[1], Meenakshi Setia [1], Bharat Prajapati[1], Santhilal Subhash [1], Mukesh Pratap Yadav[2], Subazini Thankaswamy Kosalai[1], Ananya Deshpande [1], Jagannath Kuchlyan[3], Mirco Di Marco[1], Fredrik Westerlund[4], L. Marcus Wilhelmsson [3], Chandrasekhar Kanduri [1,5] ✉ & Meena Kanduri [2,5] ✉

Although antisense transcription is a widespread event in the mammalian genome, double-stranded RNA (dsRNA) formation between sense and anti-sense transcripts is very rare and mechanisms that control dsRNA remain unknown. By characterizing the FGF-2 regulated transcriptome in normal and cancer cells, we identified sense and antisense transcripts IER3 and IER3-AS1 that play a critical role in FGF-2 controlled oncogenic pathways. We show that IER3 and IER3-AS1 regulate each other's transcription through HnRNPK-mediated post-transcriptional regulation. HnRNPK controls the mRNA stability and colocalization of IER3 and IER3-AS1. HnRNPK interaction with IER3 and IER3-AS1 determines their oncogenic functions by maintaining them in a single-stranded form. *hnRNPK* depletion neutralizes their oncogenic functions through promoting dsRNA formation and cytoplasmic accumulation. Intri-guingly, *hnRNPK* loss-of-function and gain-of-function experiments reveal its role in maintaining global single- and double-stranded RNA. Thus, our data unveil the critical role of HnRNPK in maintaining single-stranded RNAs and their physiological functions by blocking RNA-RNA interactions.

Antisense long noncoding RNAs (lncRNAs), which often overlap with the transcripts encoded from the sense strand, constitute the largest sub-class of lncRNAs[1,2]. They have been shown to regulate the expression of overlapping sense transcripts positively or negatively. Diverse antisense lncRNA-regulated transcriptional and post-transcriptional-based mechanisms have been proposed in the regulation of overlapping protein coding genes, some of which include mRNA stability, alternative splicing, and chromatin level regulation[3]. In a few instances, protein coding RNAs have also been implicated in the regulation of antisense lncRNAs[4]. In these contexts, however, the molecular mechanisms that control expression interdependence between an antisense lncRNA and its sense protein-coding partner remain unknown. Another interesting aspect is that many antisense transcripts show significant overlap with their sense counterparts[2], but there has not been much data showing RNA–RNA interaction between the over-lapping transcripts. This raises the interesting question whether there are any unique mechanisms that maintain single stranded forms of the sense and antisense transcripts to prevent RNA–RNA interactions or double-stranded RNA (dsRNA) formation between them to maintain transcript-specific functions? Thus, there are several outstanding

[1]Department of Medical Biochemistry and Cell Biology, Institute of Biomedicine, Sahlgrenska Academy, University of Gothenburg, 40530 Gothenburg, Sweden. [2]Department of Laboratory Medicine, Institute of Biomedicine, Sahlgrenska Academy, University of Gothenburg, 40530 Gothenburg, Sweden. [3]Department of Chemistry and Chemical Engineering, Chemistry and Biochemistry, Chalmers University of Technology, 41296 Gothenburg, Sweden. [4]Department of Biology and Biological Engineering, Chemical Biology, Chalmers University of Technology, 41296 Gothenburg, Sweden. [5]These authors contributed equally: Chan-drasekhar Kanduri, Meena Kanduri. ✉e-mail: kanduri.chandrasekhar@gu.se; Meena.kanduri@gu.se

questions that remain unanswered regarding the genome-wide regulation of the expression and functions of overlapping transcripts.

The FGF/FGFR signalling cascade comprises fibroblast growth factors (FGF −1 to −23) and their highly conserved transmembrane tyrosine kinase receptors (FGFR −1 to −4), which control a wide range of developmental signalling pathways with implications in cell specification during early embryonic development to organogenesis in adult organisms[5]. Over the last few years, substantial evidence has accumulated that supports aberrant FGF signalling as an underlying factor in the pathogenesis of multiple cancer types by promoting cell survival and cell proliferation via supporting angiogenesis[6,7].

FGF-2 is a prototypical example of the FGF family and its elevated levels in blood plasma have been reported in multiple cancer types[8,9]. Several studies have shown that FGF-2 is a key tumour-promoting factor, and it promotes tumour progression through downstream prosurvival signalling pathways such as, RAS-MAPK, PI3K-AKT, PLCγ, and signal transducer and activator of transcription (STAT)[10]. Consistent with its pro-survival functions, as well as its ability to induce stem cell proliferation, aberrant activation of FGF-2 signalling is associated with enhanced chemotherapy resistance[11]. Currently multiple FGFR inhibitors are being used as monotherapy or as part of combination therapies to treat therapy-resistant cancers[12]. As these inhibitors affect multiple tyrosine receptor kinases, including all FGFR receptors (1–4), it is very important to specifically target the FGF-2/FGFR pathway by understanding its downstream regulated gene circuits to minimize the side-effects of FGFR inhibitors. The functional link between FGF-2 signalling and its downstream target genes, including lncRNAs, is largely unexplored in normal and cancer cells. Thus, identifying the FGF-2 regulated gene networks in these contexts will help in devising FGF-2-based therapeutic approaches for cancer.

In this investigation, we characterize the FGF-2 regulated transcriptome in immortalized human embryonic kidney cells HEK293 and cancer cell lines such as HeLa and BT-549 and identify common and uniquely enriched FGF-2 regulated biological pathways in normal and cancer cells. By using FGF-2 induced sense and antisense transcripts IER3 and IER3-AS1 as a model system we investigate how their spatial localization, as well as RNA duplex formation between them, contribute to IER3 and IER3-AS1 oncogenic functions. Of note, we investigate the role of hnRNPK in these unique biological mechanisms.

## Results

### Characterization of FGF-2 controlled pathways in normal and cancer cell lines

To identify the FGF-2 regulated transcriptome in normal and cancer cells, we treated HEK293 and cancer cell lines like the human cervical cancer cell line HeLa, and the breast cancer cell line BT-549, with FGF-2 for 6 h and performed RNA-seq analysis. We identified several differentially expressed genes (DEGs) that were induced by FGF-2 in all the three cell lines (Fig. 1a, b). Gene Ontology (GO) analysis of the common DEGs from the three cell lines revealed several cancer-associated biological processes, such as cellular response to fibroblast growth factor stimulus, protein kinase signalling, NFKB pathway, chemokine signalling and angiogenesis (Supplementary Fig. 1a). The complete list of significant DEGs between FGF-2 treated and untreated samples and biological pathways for all the three cell lines are listed in Supplementary Data File 1. We chose to concentrate on HeLa and HEK293 cell lines for further detailed functional investigations on FGF-2 regulated gene networks and biological pathways. FGF-2 treated HEK293 and HeLa cell lines revealed several common and unique (Supplementary Fig. 1b, c, d) differentially expressed protein-coding and non-coding genes. GO analysis of cell line-specific uniquely expressed genes revealed enrichment of cell differentiation and development-associated pathways in the FGF-2 treated HEK293 cells (Supplementary Fig. 2a). On the other hand, in the FGF-2 treated HeLa cells,

overrepresentation of cancer-associated pathways such as chemotaxis, angiogenesis, cell proliferation, inflammation and apoptotic pathways was observed (Supplementary Fig. 2b), indicating different roles of FGF-2 in cancer and normal cells. GO analysis of the common FGF-2 regulated genes revealed biological pathways linked to cancer and cell differentiation (Supplementary Fig. 1a and Supplementary Data File 1). As an example, the heatmap of DEGs of cellular response to the fibroblast growth factor stimulus pathway is shown separately, which further confirms FGF-2 treatment of HEK293 and HeLa cell lines (Supplementary Fig. 2c). Enrichment of several common cancer associated pathways in the FGF-2 treated HEK293 and HeLa cells indicate a strong link between the FGF-2 signalling cascade and cancer.

Since the higher expression of FGF-2 is linked to oncogenesis, we wanted to investigate whether the treatment of normal and cancer cells with FGF-2 will lead to increase in cell proliferation. Consistent with the latter notion, FGF-2 treatment significantly enhanced cell proliferation of all the three cell lines (Fig. 1c) and also the activity of the pro-survival pathways PI3K-AKT in both normal and cancer cell lines (Fig. 1d).

### Sense and antisense transcripts IER3 and IER3-AS1 are early responders of FGF-2 treatment and control each other's expression

We obtained 126 common DEGs from the FGF-2 treated HEK293 and HeLa cells, among which four of them were lncRNAs and the remaining were protein coding genes (Supplementary Fig. 1b, c). To characterize the mechanisms underlying the FGF-2 regulated biological pathways, we chose IER3-AS1 from the four common lncRNAs considering that its sense protein coding partner gene was also induced in response to FGF-2 treatment in all the three cell lines (Fig. 1a). The RNA-seq data of FGF-2 dependent induction of IER3 and IER3-AS1 was further validated using RT-qPCR in all the three cell lines (Supplementary Fig. 2d). The basal expression of IER3 and IER3-AS1 was considerably lower in HEK293 cells compared to HeLa cells (Supplementary Fig. 2e). Interestingly, both IER3 and IER3-AS1 transcripts were activated 30 min post FGF-2 treatment and remained activated 400 min post the treatment, indicating that their induction is part of early events that occur in response to FGF-2 treatment (Supplementary Fig. 2f). In addition, both IER3 and IER3-AS1 show differential expression and strong expression correlation between them in multiple cancers from The Cancer Genome Atlas (TCGA) (Supplementary Figs. 2g, h and 3a). Thus, we chose to work on the sense and antisense pair IER3 and IER3-AS1 to understand their role in FGF-2 regulated biological pathways.

To explore the functional role of IER3 and IER3-AS1 in FGF-2 controlled functions, we downregulated IER3 and IER3-AS1 using lentiviral shRNAs in HEK293, HeLa and BT-549 cell lines. Intriguingly, downregulation of IER3 affected the expression of IER3-AS1 and vice versa in normal and cancer cell lines, indicating that they regulate each other's expression (Fig. 1e and Supplementary Fig. 3b). RNA-FISH analyses of IER3 and IER3-AS1 using RNAscope[R] probes in HeLa cells revealed that IER3 and IER3-AS1 transcripts show both nuclear and cytoplasmic localization. However, they are relatively more enriched in the nuclear compartment compared to the cytoplasmic compartment (Fig. 1f). In knockdown (KD) cells, a considerable decrease in IER3 and IER3-AS1 signals (Fig. 1f, g), as well as their intensity (Fig. 1f, h), was detected, further supporting the RT-qPCR data that IER3 and IER3-AS1 regulate each other's expression. The specificity of the IER3 and IER3-AS1 RNAscope probes was confirmed with RNAse A treated and untreated samples (Supplementary Fig. 3c, d).

### IER3 and IER3-AS1 transcripts control FGF-2 regulated pro-survival pathways

We next investigated the functional contribution of IER3 and IER3-AS1 in FGF-2 regulated functions. To this end, we analyzed RNA-seq data

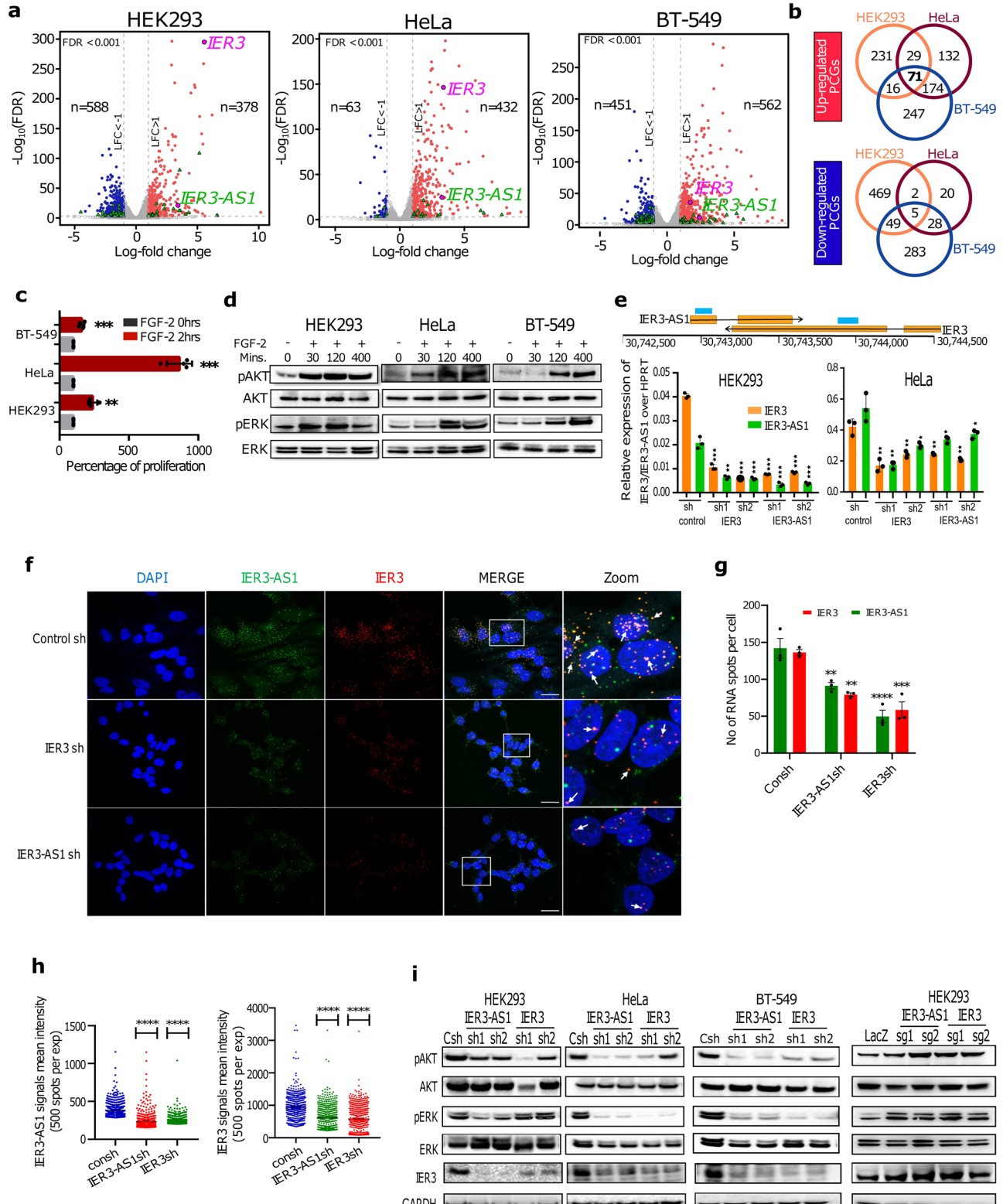

from IER3 and IER3-AS1 KD HEK293 and HeLa cells and identified several common (Supplementary Fig. 3e–g) and unique DEGs (Supplementary Fig. 3h). The complete list of DEGs from IER3 and IER3-AS1 KDs in HEK293 and HeLa cell lines along with the biological processes they take part in are listed in Supplementary Data File 2. Our data indicate that a sizeable fraction of protein coding genes and lncRNAs were commonly regulated by IER3 and IER3-AS1 in HeLa and HEK293 cell lines (Supplementary Fig. 3e–h), which is consistent with the

observation that IER3 and IER3-AS1 expression is dependent on each other's expression.

Since pro-survival pathways, such as the PI3K-AKT and MAPK/ERK pathways, were enriched in the RNA-seq data from IER3 and IER3-AS1 KD HEK293 and HeLa cell lines, we investigated the activity of these pro-survival pathways following IER3 and IER3-AS1 KD in the HEK293, HeLa and BT-549 cell lines, and overexpression in the HEK293 cell line. The intensities of pAKT and pERK, the activated forms of AKT and ERK

**Fig. 1 | FGF-2 regulated gene networks controlled by IER3 and IER3-AS1.**
**a** Volcano plots showing Log10FDR on *Y* axis and LogFC on *X*-axis of DEGs from RNA-seq data of FGF-2 untreated vs treated cell lines. Up and down regulated DEGs are shown in red and blue color dots respectively. The non-significant DEGs are represented as gray dots. Green triangles indicate significant lncRNAs and antisense RNAs. FDR threshold (=0.05) is reported as horizontal dotted gray line. FDR values are derived from DESeq2 R-package by adjusting *p*-values using Benjamini-Hochberg method. **b** Venn diagrams showing the number of common and uniquely expressed DEGs between FGF-2 treated normal and cancer cell lines. **c** Shows the percentage of cell proliferation of FGF-2 treated and untreated cells. *n* = 2 independent biological experiments and the error bars represent +/−SD. **d** Western blot showing pAKT/pERK and total AKT/ERK levels at the indicated time points in FGF-2 treated and untreated cell lines. Similar results were obtained in the independent biological replicate. **e** The schematic shows IER3 and IER3-AS1 transcripts orientation and location with respect to each other. Bar graphs show RT-qPCR analysis of IER3 and IER3-AS1 in the indicated shRNA stable knockdown (KD) samples. Significance was derived using two-sided student's *t*-test and the data represent ±SD of three biological replicates. **f** RNAscope images showing IER3 (Red) and IER3-AS1 (Green) transcripts in control and IER3/IER3-AS1 KD HeLa cells. Images at the right represents the magnified areas of the indicated white boxes from the merged images. The white arrow heads indicate colocalized dsRNA signals. DAPI was used to stain nucleus (Indicative scale bar is 50 um). *n* = 3 independent experiments. **g** Graph showing the numbers of RNA signals (spots) per cell in control sh and IER3/IER3-AS1 sh samples counted using Imaris spot detection tools. The details of quantification procedure and tools are mentioned in Supplementary Fig. 6. Total number of cells counted from three independent replicates are, control sh: *n* = 56, IER3-AS1 sh: *n* = 99 and IER3 sh: *n* = 87. Data represent mean values ± SEM and statistical significance was calculated by Two-way ANOVA. Exact *p*-values are given in the source data file. **h** Dot plots showing fluorescence's mean intensity of RNA signals in IER3-AS1sh and IER3sh cells. 500 bright intensity spots per experiment were taken for the measurement from three independent replicates. Imaris spot detection tools were used to detect the spots. Statistical significance was calculated by one-way ANOVA (Dunnett's multiple comparisons test). **i** Western blots showing the indicated protein levels in IER3 or IER3-AS1 KD or CRISPR/dcas9 activated samples in HEK293 and HeLa cell lines. GAPDH is a loading control. *n* = 2 independent experiments. Similar results were obtained in the independent biological replicate. *$p \le 0.05$; **$p \le 0.005$; ***$p \le 0.0005$, ****$p \le 0.0001$.

respectively, were significantly lower in the IER3 and IER3-AS1 KD HEK293, HeLa and BT-549 cells, whereas their levels were higher following CRISPR dependent activation of IER3 and IER3-AS1 from the endogenous loci (Fig. 1i and Supplementary Fig. 3i).

## IER3 and IER3-AS1 play a critical role in FGF-2 regulated global gene expression

To identify FGF-2 regulated genes among the IER3 and IER3-AS1 target genes, we intersected RNA-seq data from the FGF-2 treated HEK293 and HeLa cells with the RNA-seq data from the IER3 and IER3-AS1 KD HEK293 and HeLa cells (Supplementary Fig. 4a–e). In HEK293 cells, 178 FGF-2 upregulated genes (116 protein coding and 62 noncoding) and 126 FGF-2 downregulated genes (116 protein coding and 10 noncoding) were present among the downregulated and upregulated genes, respectively, from the IER3 and IER3-AS1 KDs cells (Supplementary Fig. 4a–e). However, in HeLa cells, only a few common genes (24 genes) were present in the overlaps between the FGF-2 DEGs and the DEGs from IER3 and IER3 KD cells (Supplementary Fig. 4d, e). We validated the expression status of some of the FGF-2 activated genes such as *ZFP36* and *CDKN1A* using RT-qPCR (Supplementary Fig. 4f) and Western blotting (only ZFP36: Supplementary Fig. 4g, h) in *IER3* and *IER3-AS1 KD* HEK293 and HeLa cell lines. FGF-2 dependent activation of ZFP36 and CDKN1A was dependent on IER3 and IER3-AS1, as shown using loss-of-function and/or gain-of-function experiments (Supplementary Fig. 4f–j). These results indicate that IER3 and IER3-AS1 play a crucial role in FGF-2 regulated gene expression.

## IER3 and IER3-AS1 display oncogenic features and regulate chemotaxis

As IER3 and IER3-AS1 play an important role in FGF-2 regulated prosurvival pathways, we wanted to investigate their role in cancer-associated biological processes. IER3 and IER3-AS1 KD in HEK293 and HeLa cell lines resulted in a decrease in cell proliferation and an increase in S-phase/G2 accumulation (Fig. 2a, b). An increase in apoptotic markers, such as cleaved PARP and active caspase-3, was observed only in HeLa cells compared to HEK293 cells following IER3 or IER3-AS1 KD (Fig. 2c). A decrease in colony formation as well as cell migration and invasion was observed following IER3 or IER3-AS1 KD in HeLa cells (Fig. 2d, g). Conversely, CRISPR dependent over-expression of the endogenous IER3 or IER-AS1 in HEK293 cells resulted in an increase in colony formation and cell invasion (Fig. 2d, f, g). KD of IER3 and IER3-AS1 in BT-549 cells also elicited similar effects on cell proliferation and cell cycle (Supplementary Fig. 5a, b). Consistent with the data from in vitro cell culture models, xenografts depleted with IER3-AS1 showed significant decrease in tumor volume

and considerable decrease in Ki67 staining, a cell proliferation marker (Fig. 2h–j). These results indicate that both IER3 and IER3-AS1 possess putative oncogenic properties in HEK293, HeLa and BT-549 cells.

We noted that there was an overrepresentation of chemokine related pathways in our RNA-seq data from the FGF-2 treated HEK293, HeLa and BT-549 cells and in the IER3 and IER3-AS1 KD HEK293 and HeLa cells (Supplementary Fig. 1a and Supplementary Data File 2). Furthermore, some of the CC and CXC family chemokines were abundantly expressed in the cancer cell lines HeLa and BT-549 compared to the HEK293 cells (Supplementary Fig. 5c). These chemokines showed significant induction following FGF-2 treatment (Supplementary Fig. 5d) and were downregulated in the IER3 or IER3-AS1 KD cells (Supplementary Fig. 5e, f). We validated FGF-2, IER3 and IER3-AS1 dependent regulation of the chemokines by RT-qPCR (Supplementary Fig. 5g, h). In addition, xenografts lacking IER3-AS1 showed lower expression of the validated chemokines (Supplementary Fig. 5i). These observations indicate that FGF-2, IER3 and IER3-AS1 regulated chemokines constitute pro-tumorigenic signature and take part in tumor progression.

We next investigated the significance of FGF-2/IER3-AS1 controlled chemokine pathways in cell migration using the μ-slide chemotaxis assay, which measures the chemotactic behavior of cells in 3D gel matrices. Considering that chemokines play a crucial role in establishing the tumor microenvironment through the recruitment of immune cells, we wanted to test whether the growth factor induced chemokines control the migratory behavior of human peripheral blood mononuclear cells (PBMCs) using μ-slide chemotaxis assay. In this assay, we measured the effect of the FGF-2 treated control and the IER3-AS1 KD HeLa cells on the migration potential of PBMCs. We found that PBMCs showed more migratory behavior towards FGF treated control HeLa cells compared to IER3-AS1 KD cells (Fig. 2k, l). These results, taken together, indicate that there is a strong functional relationship between FGF-2 and IER3-AS1 in the regulation of chemokine pathways.

We investigated the significance of FGF-2/IER3-AS1 regulated CC and CXC chemokines in the regulation of cancer associated biological processes such as cell proliferation, pro-survival pathway regulation and cell invasion. For that we have selected CCL20 and CXCL1 and validated their expression dependence on FGF-2/IER3/IER3-AS1 regulatory axis (Supplementary Fig. 5j). KD of these two chemokines resulted in significant decrease in cell proliferation, activity of pro-survival pathways, cell migration/invasion and cell colonization (Supplementary Fig. 5k–q). These results highlight the role of the CC and CXC chemokines in the regulation of FGF-2/IER3-AS1 cancer-associated biological actions.

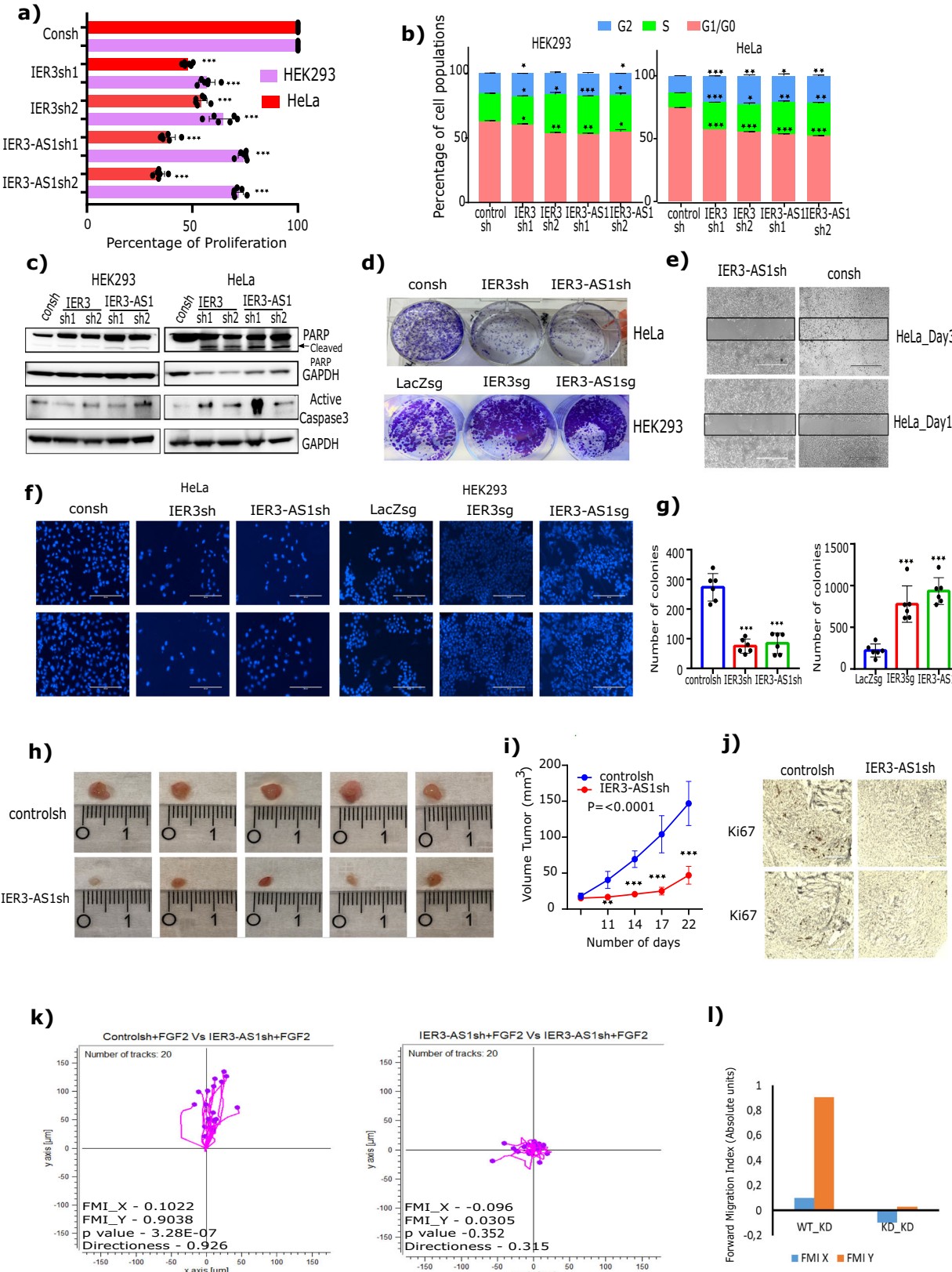

## IER3 and IER3-AS1 show colocalization and control each other's stability

IER3 and IER3-AS1 are sense-antisense transcripts encoded from a convergent pair of genes, which share a 325 bp overlapping region in their downstream exon 2 region (Fig. 3a). Using the bio.tools software[13], we identified the IER3 and IER3-AS1 interacting regions with the lowest binding free energy in the overlapping region (Fig. 3b). Consistent with the in silico predictions, we confirmed the interaction between the IER3 and IER3-AS1 transcripts by using the chromatin oligo affinity precipitation (ChOP) technique[14], wherein we pulldown IER3-AS1 using biotinylated antisense oligonucleotides. This IER3-AS1-based ChOP assay detected both IER3 and IER3-AS1

**Fig. 2 | IER3 and IER3-AS1 harbor oncogenic properties. a** Cell viability assay showing the percentage of cell proliferation in IER3 and IER3-AS1 stable KD HeLa and HEK293 cells. **b** Distribution plots showing the percentage of cell populations of G0/G1, S and G2 phases of cell cycle in IER3 and IER3-AS1 stable KD HeLa and HEK293 cells. Fig. **a, b**: values represent mean ± SD of two biological experiments. Significance was calculated using two-sided Student's *t*- test. **c** Western blot showing the expression levels of different apoptotic proteins probed with the indicated antibodies. **d** Colony forming efficiency of IER3 and IER3-AS1 stable KD HeLa cells (upper panel) and CRISPR/dcas9 activated IER3 and IER-AS1 HEK293 cells (lower panel). **e** Wound healing or scratch assay showing cell migration efficiency of control sh and IER3-AS1sh stable KD HeLa cells. Similar results were obtained from two independent experiments for (**c**), (**d**) and (**e**). **f** Transwell invasion assay showing the invasive potential of IER3 and IER3-AS1 stable KD HeLa cells and stable overexpressing (OE) (CRISPRa) HEK293 cells. **g** Bar graphs show quantification of the invasive cells shown in Figure (**f**). The invasive cells were counted using ImageJ software and error bars represents ± SD from two

independent experiments. **h** Representative images of tumors (*n* = 5) depicting the tumor size of xenografts that were developed by subcutaneously injected HeLa cells stably transduced with control sh or IER3-AS1 shRNA. Tumors were harvested after 25 days post injection. **i** The graph showing the difference in the volume of tumors between HeLa control sh mice and HeLa IER3-AS1sh mice group (each group contains 5 mice). Error bars represents ± SD and significance was calculated using Student *t*-test. **j** Representative images of the Ki67 staining on the cross sections of xenografts from Fig. 2h. The data shown here is from two individual mice tumors. **k** Migration trajectory graphs showing the trajectory of PBMCs in response to FGF-2 treatment of HeLa control sh and HeLa IER3-AS1 stable KD cells. The migration assay was performed using 3D cell migration chemotaxis from ibidi. The forward migration index value (FMI) for both *X* and *Y* axis, the Rayleigh test (*p*-value) are mentioned for each plot. Similar results were obtained in two independent experiments. **l** The graph showing the FMI values from Fig. 2k for both HeLa control sh and HeLa IER3-AS1 KD cells. *$p \le 0.05$; **$p \le 0.005$; ***$p \le 0.0005$.

transcripts (Fig. 3c), indicating that these sense and antisense transcripts interact with each other. Interaction between the IER3 and IER3-AS1 transcripts was also evident in RNAscope experiments of HeLa and HEK293 cells as well as in FGF-2 treated HeLa cells (Fig. 3d, e). Upon FGF-2 induction, we found an increase in IER3 and IER3-AS1 levels and their co-localization (Fig. 3e). Furthermore, the stability of the IER3 and IER3-AS1 transcripts was analyzed by KD of one transcript and quantifying the other transcript after different time points of incubation with ActinomycinD. The results showed that the stability of the IER3 and IER3-AS1 transcripts were dependent on each other, as KD of one transcript affected the stability of the other and vice versa (Fig. 3f). These results imply that RNA−RNA interaction between these transcripts may promote their stability and function.

### Nuclear enriched protein HnRNPK interacts with IER3-AS1 and IER3

Considering the interdependence of IER3-AS1 and IER3 expression on each other, one of the important questions that arises from these observations is whether IER3 and IER3-AS1 regulate each other at the post-transcriptional level? To address this very important question, we first characterized the IER3-AS1 interacting proteome using ChOP-mass spectrometry and identified nearly 50 IER3-AS1 interacting proteins that show two-fold enrichment over a LacZ control (Supplementary Data File 3). Among the IER3-AS1 interacting proteins, we selected hnRNPC and its functional interactor HnRNPK, from the top candidate list to understand their potential role in post-transcriptional regulation of IER3 and IER3-AS1 (Supplementary Fig. 5r). KD of *hnRNPK* but not *hnRNPC* resulted in significant activation of both IER3 and IER3-AS1, indicating a strong functional connection between *hnRNPK*, IER3 and IER3-AS1 expression (Fig. 3g, h). Additionally, RNA sequencing analysis showed a significant increase in levels of IER3 and IER3-AS1 expression following *hnRNPK* KD in HeLa cells (Fig. 3i). We validated the interaction of HnRNPK with IER3-AS1 using RNA immunoprecipitation (RIP). Interestingly, IER3, which we considered as a negative control along with GAPDH and HPRT, was also present in the HnRNPK RIP along with ZFP36, a positive control for the experiment (Fig. 3j). This indicates that HnRNPK interacts with both sense IER3 and antisense IER3-AS1 transcripts. In addition, we have observed colocalization of HnRNPK with IER3-AS1 in Immuno-RNA-FISH experiments (Fig. 3k), which further corroborates IER3-AS1 and HnRNPK interaction. Supporting our data on HnRNPK interaction with IER3 and IER3-AS1, HnRNPK RIP-sequencing analysis of a published dataset[15] (GSE122327) from HEK293 cells revealed HnRNPK peaks at the IER3 and IER3-AS1 locus (Fig. 3l, m), which further reinforces the specificity of the HnRNPK interaction with IER3-AS1 and IER3.

### HnRNPK controls RNA−RNA interactions between IER3 and IER3-AS1 and their sub-cellular localization

As HnRNPK showed interaction with the sense and antisense transcripts, we next investigated the significance of HnRNPK in the regulation of RNA-RNA interactions between IER3 and IER3-AS1 transcripts and their functions. The RNA-Scope experiments revealed that in control HeLa cells, IER3-AS1 and IER3 transcripts were present as an individual signals or in colocalization with each other in both nuclear and cytoplasmic compartments (Fig. 4a). Overall, there was a preferential nuclear enrichment of RNA-FISH signals in the HeLa and HEK293 cell lines (Figs. 1f, 3d and 4a). This was evident in RT-qPCR analysis of the nuclear and cytoplasmic RNA fractionations (Fig. 4b). In *hnRNPK* KD cells, however, we found an increase in the expression of both IER3 and IER3-AS1, as evident in RNA-seq, RT-qPCR and RNAscope experiments (Fig. 3h, i and 4a, c). Intriguingly, individual RNA-FISH signals of IER3-AS1 showed a significant decrease, whereas there was a robust increase in its colocalized signals with IER3 in the *hnRNPK* KD cells (Fig. 4a, d, e), indicating that HnRNPK controls RNA−RNA interactions between IER3-AS1 and IER3. The methodology and the exact calculations used for analyzing the 3D image dots in RNAscope experiments is shown in Supplementary Fig. 6a, b. We found significant accumulation of the colocalized signals in the cytoplasmic compartment compared to the nuclear compartment in the *hnRNPK* KD cells (Fig. 4a). We validated the cytoplasmic accumulation of IER3 and IER3-AS1 by RT-qPCR and RNA-seq in the *hnRNPK* KD cells (Fig. 4b, f). Thus, our work uncovers an hnRNPK-dependent pathway which determines individual transcript functions by regulating RNA−RNA interactions.

To identify the transcripts that are accumulated in the cytoplasm of *hnRNPK* KD cells, RNA from the cytoplasmic and nuclear compartments of control and *hnRNPK* KD cells was sequenced. 666 transcripts, including IER3 and IER3-AS1, showed more cytoplasmic enrichment in *hnRNPK* KD cells compared to control cells, whereas only 226 transcripts showed nuclear enrichment (Fig. 4f). Interestingly, both IER3 and IER3-AS1 were shown to be differentially enriched only in the cytoplasmic RNA but not in the nuclear RNA (Fig. 4f). Previously, HnRNPK has been shown to retain RNA in the nuclear compartment through interacting with CCTCC motifs[16]. Both IER3-AS1 and IER3 transcripts harbor such motifs in the IER3/IER3-AS1 overlapping region as detected using FIMO (Find individual motif occurrence) analysis[17] (Supplementary Fig. 7a). Taken together, these observations indicate that HnRNPK plays an important role in the formation of RNA−RNA interactions between IER3 and IER3-AS1. This observation also implies that HnRNPK may be involved in keeping these transcripts in single stranded form in the nuclear and cytoplasmic compartments to maintain their functions and its depletion leads to increased RNA-RNA interactions between IER3 and IER3-AS1 and cytoplasmic accumulation.

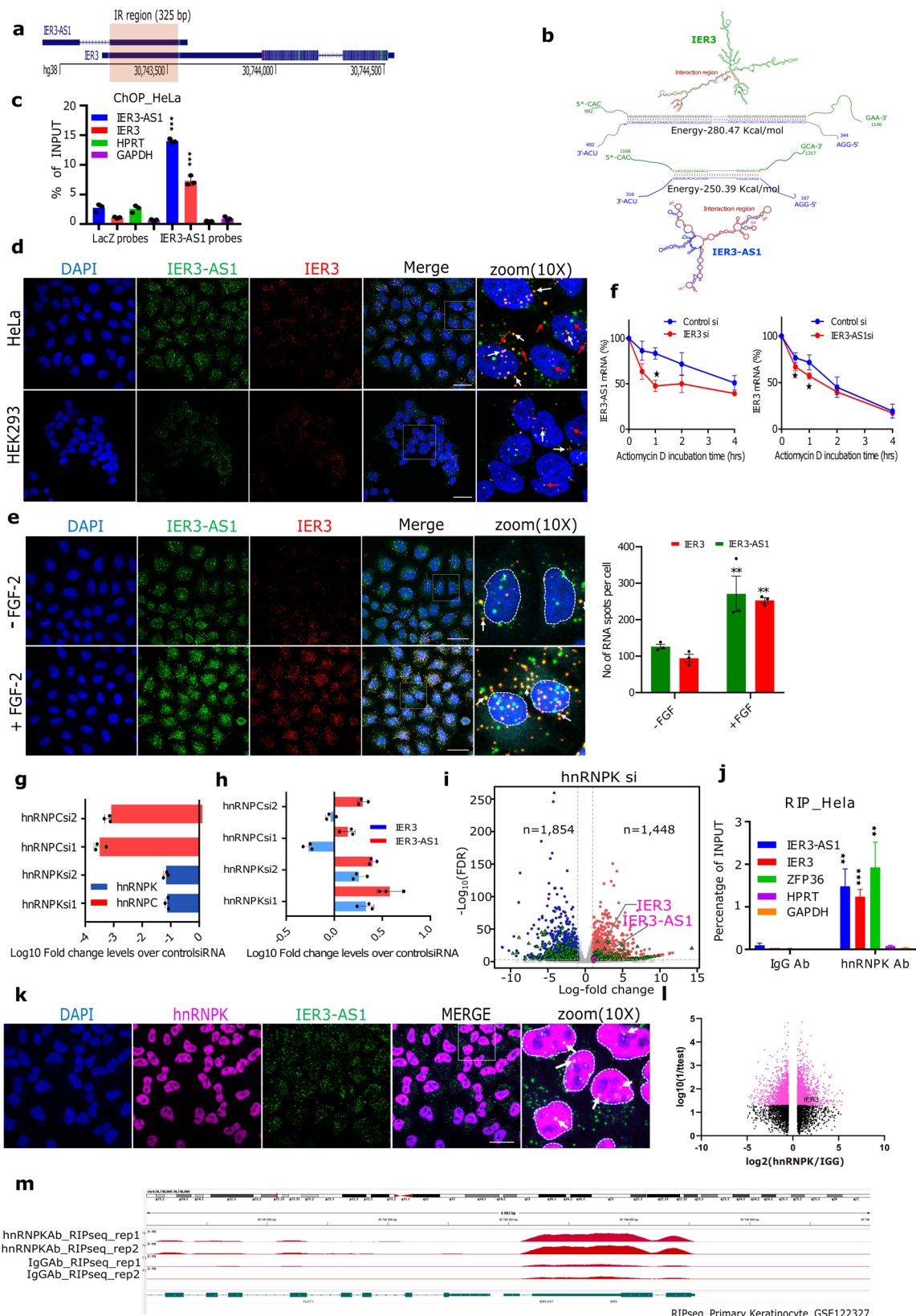

**RNA–RNA interactions between IER3 and IER3-AS1 correlate with increase in post-transcriptional gene expression of IER3**

Since we observed more cytoplasmic accumulation as well as colocalization of IER3 and IER3-AS1 in hnRNPK KD cells, we investigated whether IER3 and IER3-AS1 colocalization increases their stability. IER3 and IER3-AS1 half-lives were significantly increased following *hnRNPK*

KD (Fig. 4g), indicating that the colocalization of IER3 and IER3-AS1 increases their stability, potentially through the formation of RNA–RNA interactions. Since the colocalization correlates with an increase in the stability of IER3 and IER3-AS1 transcripts, we next asked the question whether their colocalization is linked to an increase in IER3 expression at the post-transcriptional level. To address this, we

**Fig. 3 | Post-transcriptional control IER3 and IER3-AS1 by HnRNPK. a** The schematic of overlapping region (IR) between IER3 and IER3-AS1 transcripts. **b** RNA secondary structures and the predicted RNA–RNA interactions of IER3 and IER3-AS1. IER3 and IER3-AS1 specific sequences are shown in green and blue color respectively. Red color highlight the IR region. **c** RT-qPCR analysis of chromatin oligo affinity precipitation (ChOP) assay performed using 8 biotinylated IER3-AS1-specific antisense oligos. LacZ-specific antisense oligos were used as negative pull-down control. **d, e** RNAscope images showing IER3 (red) and IER3-AS1(green) transcripts in HeLa and HEK293 cells (**d**) and in HeLa cells treated with FGF-2 (**e**). Images at the right are magnified areas of the indicated white boxes from the merged images. The white and red arrow heads indicate colocalized dsRNA and ssRNA signals, respectively. Indicative scale bar on the images is 50um. The graph in Fig. 3e shows the number of IER3-AS1 and IER3 RNA signals upon FGF treatment, quantified as RNA spots per cell using Imaris spot detection tools. A total, -FGF: $n = 116$, and +FGF: $n = 101$ cells were counted from 3 independent experiments and data represent mean ± SEM. The significance was calculated using Two-way ANOVA. **f** RT-qPCR showing IER3-AS1 and IER3 transcripts at the indicated time points in actinomycin D treated IER3 or ER3-AS1 KD cells. **g, h** RT-qPCR showing Log10FC values of *hnRNPK* and *hnRNPC* (**g**) and IER3 and IER3-AS1 (**h**) in *hnRNPK* and *hnRNPC*

KD HeLa cells. **I** Volcano plot showing HeLa cell RNA-seq data of control siRNA vs *hnRNPK* siRNA. Upregulated, downregulated and non-significant DEGs genes are shown in red, blue and grey dots, respectively. The dotted lines indicate the FDR threshold (=0.05). FDR values are derived from DESeq2 R package by adjusting *p*-values using Benjamini-Hochberg method. **j** RT-qPCR analysis of RNA immuno-precipitation (RIP), performed in HeLa cells. ZFP36 gene was a positive control and HPRT and GAPDH were negative controls for the pull-down. Data were normalized to the input RNA and plotted as the percentage of input. $n = 3$ independent replicates and error bars indicate +/− SD. Significance was calculated using 2-sided Student's-*t*-test for C, F, G, H and J. **k** Immunofluorescence (IF)-RNAscope images show IER3-AS1 (green) colocalization with HNRNPK immunostaining (magenta) in HeLa cells. Magnified areas from the merged as indicated in white boxes are shown on right-side. The co-localized signals are indicated by white arrows. Indicative scale bar on the images is 50 um. **l, m** Volcano plot showing the DEGs (**l**) and HnRNPK binding peaks over the IER3/IER3-AS1 locus (**m**) from the downloaded published RIP-sequencing dataset (GSE122327). The HnRNPK RIP data was presented over IgG. Significant and non-significant genes are shown in pink and black color, respectively. For Fig. (**c, e, f** and **j**): *$p \le 0.05$; **$p \le 0.005$; ***$p \le 0.0005$.

---

performed immune-RNA-FISH on *hnRNPK* KD cells with IER3 and IER3-AS1 RNAscope probes and a ribosome-specific antibody RSP3. We found that more IER3 and IER3-AS1 signals were colocalized with ribosome staining in *hnRNPK* KD cells, indicating active translation of IER3 (Fig. 4h, i). Furthermore, higher IER3 levels were detected in the immunoblots from *hnRNPK* KD cells (Fig. 4j).

### HnRNPK determines IER3-AS1 and IER3 oncogenic functions through preventing their colocalization

Our data clearly demonstrate that IER3-AS1 with its nuclear and cytoplasmic localization exhibit oncogenic properties. IER3-AS1 is localized in both single and double stranded form. This raises the important question of whether IER3-AS1 possesses oncogenic properties in its single or double stranded form? IER3-AS1 sub-cellular distribution is controlled by HnRNPK as *hnRNPK* depletion led to more cytoplasmic accumulation and colocalization with IER3. More importantly, our data demonstrate that despite an increase in IER3 and IER3-AS1 in *hnRNPK* KD cells, we did not notice any increase in oncogenic properties (Supplementary Fig. 7b–d), indicating that IER3 or IER3-AS1 per se do not possess oncogenic functions, but rather that the interaction with HnRNPK determines their oncogenic properties. Moreover, HnRNPK has been shown to play a critical role in tumor progression by regulating metastasis and angiogenesis[18]. Based on these observations, we hypothesize that HnRNPK interaction with single stranded IER3-AS1 and IER3 is crucial in determining their oncogenic functions and the loss of HnRNPK neutralizes the oncogenic functions of IER3-AS1 and IER3 by promoting RNA-RNA interactions between them. To investigate whether HnRNPK shows preference for single-stranded IER3-AS1 and IER3 or double-stranded IER3-AS1/IER3, we incubated HeLa cell extracts with biotinylated single stranded IER3, IER3-AS1 and IER3/IER3-AS1 RNA duplex as well as IER3-AS1 harboring mutations at the CCTCC (CCTC to AAGA motifs). Immunoblot analysis of the streptavidin pulldowns revealed that IER3-AS1 displayed higher HnRNPK interaction compared to IER3 and IER3/IER3-AS1 duplex RNA. However, mutated IER3-AS1 showed the least interaction with HnRNPK. These results indicate that IER3-AS1 interacts specifically with HnRNPK and this specific interaction is mediated through CCTCC motifs (Fig. 5a). Additionally, using a fluorescence titration experiment (Fig. 5b and Supplementary Fig. 7e) we demonstrated that the single-stranded IER3 and IER3-AS1 transcripts have significantly higher binding constants compared to the annealed double-stranded IER3/IER3-AS1. This further supports our hypothesis that HnRNPK binds stronger to IER3 and IER3-AS1 single-stranded RNA compared to the dsRNA. To further explore the role of HnRNPK in determining the IER3-AS1 in single or double strand form, we scored IER3-AS1 specific RNAscope signals after constructing 3D images of RNA-FISH using

IMARIS (Supplementary Fig. 6). We found that *hnRNPK* KD causes a significant decrease in IER3-AS1 specific signals and a robust increase in its colocalized signals with IER3 in the *hnRNPK* KD cells, further suggesting that HnRNPK contributes to IER3-AS1 and IER3 oncogenic functions by keeping them in single stranded form (Fig. 4a–e).

We were next interested in investigating the cancer-specific hall-marks of *hnRNPK* KD cells where both IER3 and IER3-AS1 were upregulated. We found a significant increase in apoptosis and a decrease in proliferation of *hnRNPK* KD cells (Supplementary Fig. 7b–d). The increased apoptosis in *hnRNPK* KD cells could be due to increased dsRNA formation. To examine this, we analyzed downstream dsRNA sensors such as RNA helicase RIG-I and MDA-5 (melanoma differentiation-associated gene 5)[19] which recognize short and long cytoplasmic dsRNAs, respectively, and induce apoptosis in a dsRNA dependent manner. Interestingly, we found increased levels of RIG-1 in *hnRNPK* KD samples (Fig. 4k), but not MDA-5. Furthermore, we found a robust increase in apoptosis and decrease in cell proliferation of HeLa cells transfected with IER3-AS1/IER3 duplex RNA compared to single stranded IER3-AS1, which is in line with *hnRNPK* KD cells (Supplementary Fig. 7f–h).

Since the overlapping (IR) region of the IER3 and IER3-AS1 transcripts harbors a maximum number of CCTCC motifs and the single stranded sense (IER3) and antisense (IER3-AS1) RNA forms of IR region showed a higher binding constant for HnRNPK compared to the dsRNA form, we generated a 523 bp targeted deletion encompassing the IR region (Supplementary Fig. 7i–k). The IR deletion significantly reduced the level of IER3 and IER3-AS1 RNA levels (Fig. 5c) and also phenocopies shRNA dependent KD of IER3-AS1 and IER3 in terms of its effect on cell proliferation and cell cycle regulation (Supplementary Fig. 7l–n). These observations are further supported by a considerable overlap among the DEGs of IR deletion vs. IER3-AS1sh KD (60% overlap) and IR deletion vs. IER3sh KD (58% overlap) (Supplementary Fig. 7o). Besides, to check the specificity of HnRNPK binding to the IR region at the CCTCC motifs, we generated serial deletions of the IER3-AS1 encompassing different number of CCTCC motifs and also IER3-AS1 containing mutations at all the CCTCC motifs. Overexpression of the IER3-AS1 deletion lacking all CCTCC motifs or CCTCC mutated IER3-AS1 in HeLa wild-type, HeLa IR deletion cells did not rescue the cell proliferation phenotype of the IR deletion cell line when compared to the cell lines overexpressing wild type IER3-AS1 or IER3-AS1 IR region (Fig. 5d). Similar results were also observed in HeLa IER3-AS1sh cells (Supplementary Fig. 8a). The expression levels of the various deletions and CCTCC mutant IER3-AS1 are shown in Supplementary Fig. 8b, c. These results indicate that HnRNPK contributes to IER3-AS1 and IER3 oncogenic functions by maintaining them in single stranded form by interacting

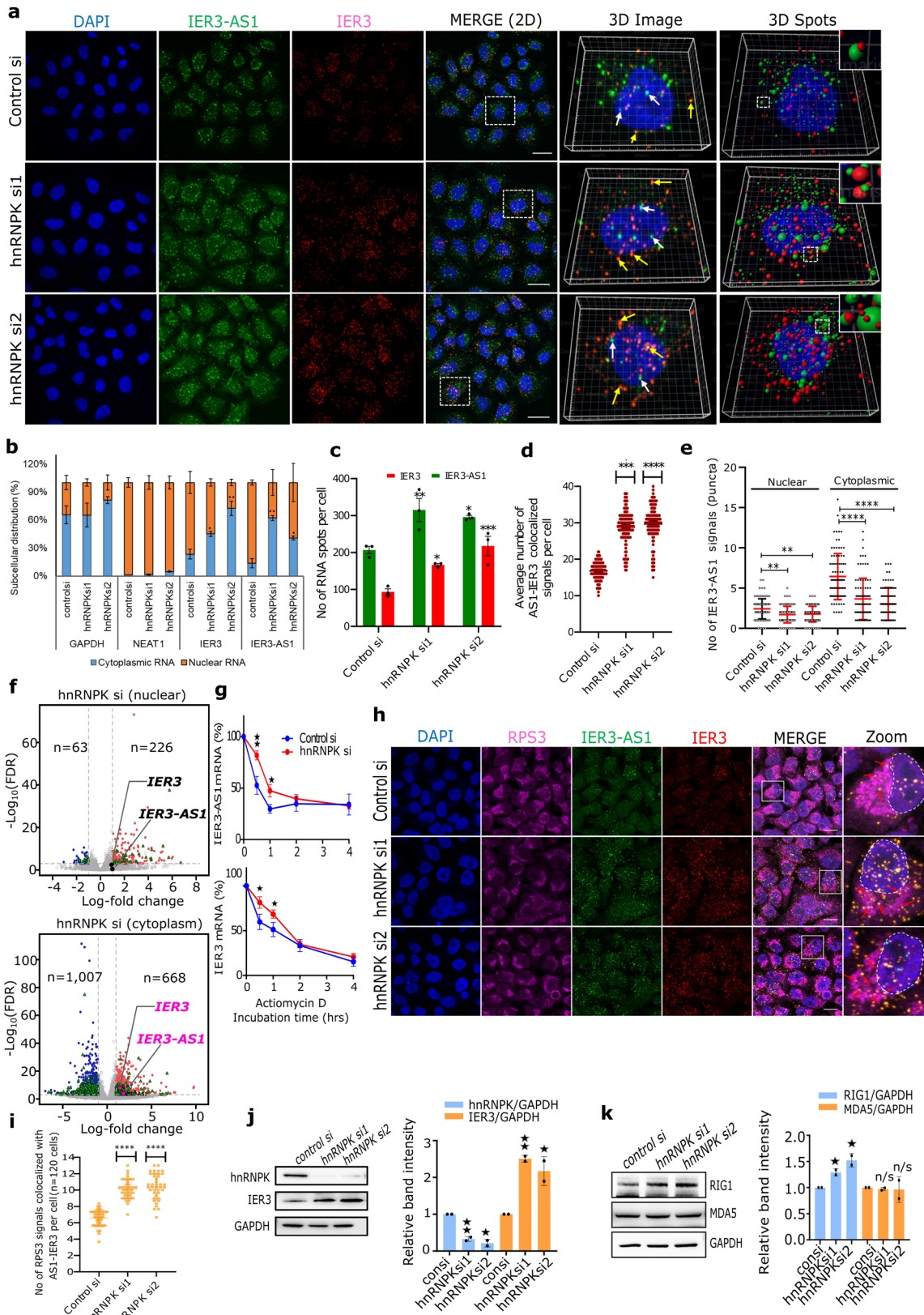

with CCTCC motifs and that this interaction is crucial for blocking RNA–RNA interactions between IER3 and IER3-AS1.

## HnRNPK controls global single and double-strand RNA

As our data points towards increased dsRNA formation between IER3 and IER3-AS1 in the *hnRNPK* KD cells, we wanted to further confirm

this using an antibody J2 that specifically detects dsRNA. We first performed immuno-RNA-FISH using IER3 and IER3-AS1 RNAscope probes followed by immunostaining with J2. We found that the IER3 and IER3-AS1 colocalized signals were part of dsRNA staining and that there was a significant increase in J2 signals that overlap with the colocalized IER3-AS1 and IER3 signals (Fig. 5e). In addition, IER3 and

**Fig. 4 | HnRNPK determines RNA–RNA interactions and sub-cellular localization. a** RNAscope images of IER3 (red) and IER3-AS1 (green) in hnRNPK KD HeLa cells. The right-side panels show the 3D reconstructed images (Imaris) of the merged 2D image indicated by white dotted box. IER3-AS1- white arrows and IER3 and IER3-AS1 dsRNA - yellow arrows. Green spheres (IER3-AS1) and red spheres (IER3) shown in the 3D images are detected using spot detection tool. Upper right corner inset shows magnified co-localized spots. Indicative scale bar is 50 um. *n* = 3 independent experiments. **b** RT–qPCR data showing the nuclear and cytoplasmic distribution of IER3 and IER3-AS1. GAPDH and NEAT1 were cytoplasmic and nuclear RNA controls, respectively. **c** Quantification of IER3-AS1 and IER3 RNA signals using Imaris spot detection tools. Cells counted-control: *n* = 81, siRNA1: *n* = 74 and siRNA2: *n* = 75. The statistical significance was calculated by two-way ANOVA (Sidak's multiple comparisons test). **d** Dot plots showing the average number of IER3-AS1/IER3 colocalized signals per cell (*n* = 150 cells). The statistical significance was calculated by one-way ANOVA (Kruskal–Wallis test). **e** Graph showing the number of non-colocalized IER3-AS1 signals (Fig. 4a: green dots pointed with white arrows) in HeLa cells (*n* = 90 cells from 3 independent experiments). The statistical significance was calculated by one-way ANOVA (Kruskal–Wallis test). f) Volcano plots showing Log2FC values of DEGs of RNA-seq data from nuclear (upper panel) and cytoplasmic

(lower panel) fractions. Upregulated DEGs - red dots; down regulated DEGs - blue dots; non-significant DEGs - gray dots; lncRNAs: green triangles and FDR threshold (=0.05) - horizontal dotted gray line. FDR values were derived from DESeq2 R-package by adjusting *p*-values using Benjamini-Hochberg method. IER3 and IER3-AS1 were indicated as roseate dots and black dots. **g** RT–qPCR showing IER3-AS1 and IER3 levels in hnRNPK KD cells at the indicated time points of actinomycin D incubation. **h** IF-RNAscope images showing RPS3 (magenta), IER3 (red) and IER3-AS1 (green) signals in HeLa cells. The panel to the right shows the zoomed image of the merged panel indicated by white box. Red arrows show co-localized RPS3/IER3/IER3-AS1 signals. The dotted white lines mark the nucleus. Indicative scale bar is 50 um. **i** Dot plots showing the average number of colocalized RPS3/IER3-AS1/IER3 signals per cell (*n* = 120 cells from 3 independent replicates). The statistical significance calculated using one-way ANOVA (Kruskal–Wallis test). **j** Western blots show HNRNPK and IER3 levels in *hnRNPK* KD cells. Histograms shows the relative band intensities from the western blot. **k** Western blots showing the indicated protein levels in *hnRNPK* KD cells. Histograms showing the relative band intensities from the western blot. The *p* values were calculated using two-sided Student's *t*-test and data are presented ± SD from two independent replicates for (**b**, **g**, **j** and **k**). *$p \leq 0.05$; **$p \leq 0.005$; ***$p \leq 0.0005$, ****$p \leq 0.0001$.

IER3-AS1 transcripts were enriched in RIP assays performed with J2 antibody on RNAseA treated RNA from *hnRNPK* KD samples (Fig. 5f). We also observed enrichment of IER3 and IER3-AS1 transcripts in the total RNA that was digested with RNase A and analyzed enrichment of dsRNA against GAPDH (ssRNA) (Fig. 5g). *ERVL* and *MTL2B4* genes were used as positive controls for dsRNA[20,21]. Collectively, these results indicate that the loss of *hnRNPK* results in *IER3* and *IER3-AS1* dsRNA formation.

As *hnRNPK* KD led to the colocalization of IER3 and IER3-AS1 through dsRNA formation, we investigated whether HnRNPK contributes to the global dsRNA formation in HeLa cells. When we analyzed dsRNA formation following *hnRNPK* KD using the dsRNA-specific antibody J2, we found a significant increase in J2 specific staining in the cytoplasmic compartment of *hnRNPK* KD cells (Fig. 5h) and this staining was sensitive to RNAse III treatment (Supplementary Fig. 8d). We further confirmed dsRNA formation in *hnRNPK* KD cells using RNA dot blots (Supplementary Fig. 8e). Interestingly, a spike in dsRNA-specific J2 signals was also observed in the breast cancer (BT-549) and lung adenocarcinoma (A549) cell lines following *hnRNPK* KD (Fig. 6a, b). Similarly, an increase in the colocalization of IER3 and IER3-AS1 transcripts was also observed in both A549 and BT-549 cells (Fig. 6c–e) as well as an increase in IER3 protein levels, as detected by immunoblot in A549 cells, following *hnRNPK* KD (Fig. 6d). We then investigated if global dsRNA formation is only specific for HnRNPK or if the other IER3-AS1 interacting proteins also contribute to global dsRNA formation. To this end, we selected another IER3-AS1 interacting protein from the top 50 protein list of ChOP-mass spec called SFN. SFN downregulation using siRNA in both HeLa and A549 cells did not affect the levels of IER3 and IER3-AS1 and also not global dsRNA levels (Supplementary Fig. 8f, g).

To further reinforce the role of HnRNPK in the regulation of global dsRNA formation, we overexpressed *hnRNPK* in the A549 cell line and analyzed dsRNA formation. We found that *hnRNPK* overexpression drastically reduced dsRNA formation in A549 cells (Fig. 7a, b). To investigate whether *hnRNPK* overexpression blocks dsRNA formation between IER3 and IER3-AS1, we analyzed the levels of single- and double-stranded RNA forms of IER3 and IER3-AS1 in the *hnRNPK* overexpression cells using RNAscope. We found that the hnRNPK overexpressing cells showed a significant decrease in IER3 and IER3-AS1 levels and, intriguingly, there was more enrichment of single-stranded IER3 and IER3-AS1 as opposed to the control cells, which harbored more colocalized IER3/IER3-AS1 dsRNA (Fig. 7c–e). In line with this data, immunoblots detected lower IER3 levels upon *hnRNPK* overexpression (Fig. 7b), indicating that HnRNPK plays a critical role in the regulation of global RNA–RNA interactions.

## Discussion

Our FGF-2 based transcriptomic analysis of immortalized human embryonic kidney cells HEK293 and cancer cell lines HeLa and BT-549 identified gene regulatory networks that execute their biological functions in normal and neoplastic conditions. Of note, our work has uncovered FGF-2 induced sense and antisense transcripts IER3 and IER3-AS1, encoded from a convergent pair of genes, as crucial entities in FGF-2 dependent cancer associated biological pathways, such as pro-survival and chemokine regulated cell migratory behavior. Interestingly, regulation of FGF-2-induced expression of IER3 and IER3-AS1 occurs post-transcriptionally through HnRNPK dependent spatially controlled RNA-RNA interactions. Our data suggests that HnRNPK interactions with IER3 and IER3-AS1 determine their oncogenic functions through retaining them in single stranded form and the loss of HnRNPK results in the accumulation of both the transcripts in the cytoplasmic compartment, but in dsRNA form.

Our data highlights HnRNPK dependent unique gene expression regulation that occurs at the post-transcriptional level. We show that HnRNPK binding is crucial for maintaining IER3 and IER3-AS1 turnover, as its KD results in an increase in the stability as well as the levels of both the transcripts. These observations are consistent with HnRNPK role in post-transcriptional gene regulation[22]. HnRNPK, a member of the nuclear enriched poly (C)-binding protein (PCBP) family, was shown to be necessary for pre-mRNA metabolism[23,24] and it also participates in stabilizing mRNA through binding to the 3' untranslated region (3' UTR)[25]. Stabilization as well as cytoplasmic colocalization of IER3 and IER3-AS1 transcripts following *hnRNPK* KD, indicate that HnRNPK utilizes a mode of action in the post-transcriptional gene regulation of IER3 and IER3-AS1 where HnRNPK through its specific interactions with CCTCC motifs in the overlapping IR region blocks RNA-RNA interactions between IER3 and IER3-AS1. In line with the latter notion, our RNAscope experiments clearly demonstrated that *hnRNPK* loss leads to increased colocalization of IER3 and IER3-AS1 whereas its overexpression resulted in loss of their colocalization and enrichment of single-stranded IER3 and IER3-AS1. Thus, these results suggest a hitherto unknown role of hnRNPK in maintaining single stranded forms of IER3 and IER3-AS1 to maintain their oncogenic functions.

One of the most interesting observations of the present investigation is robust colocalization of IER3 and IER3-AS1 transcripts. We supported this notion by in silico predictions using the bio.tools software, which identified IER3 and IER3-AS1 interacting regions with the lowest binding free energy. This in silico prediction data was further validated using multiple complementary technologies such as IER3-AS1 ChOP, HnRNPK RIP pulldowns and immuno RNAscope.

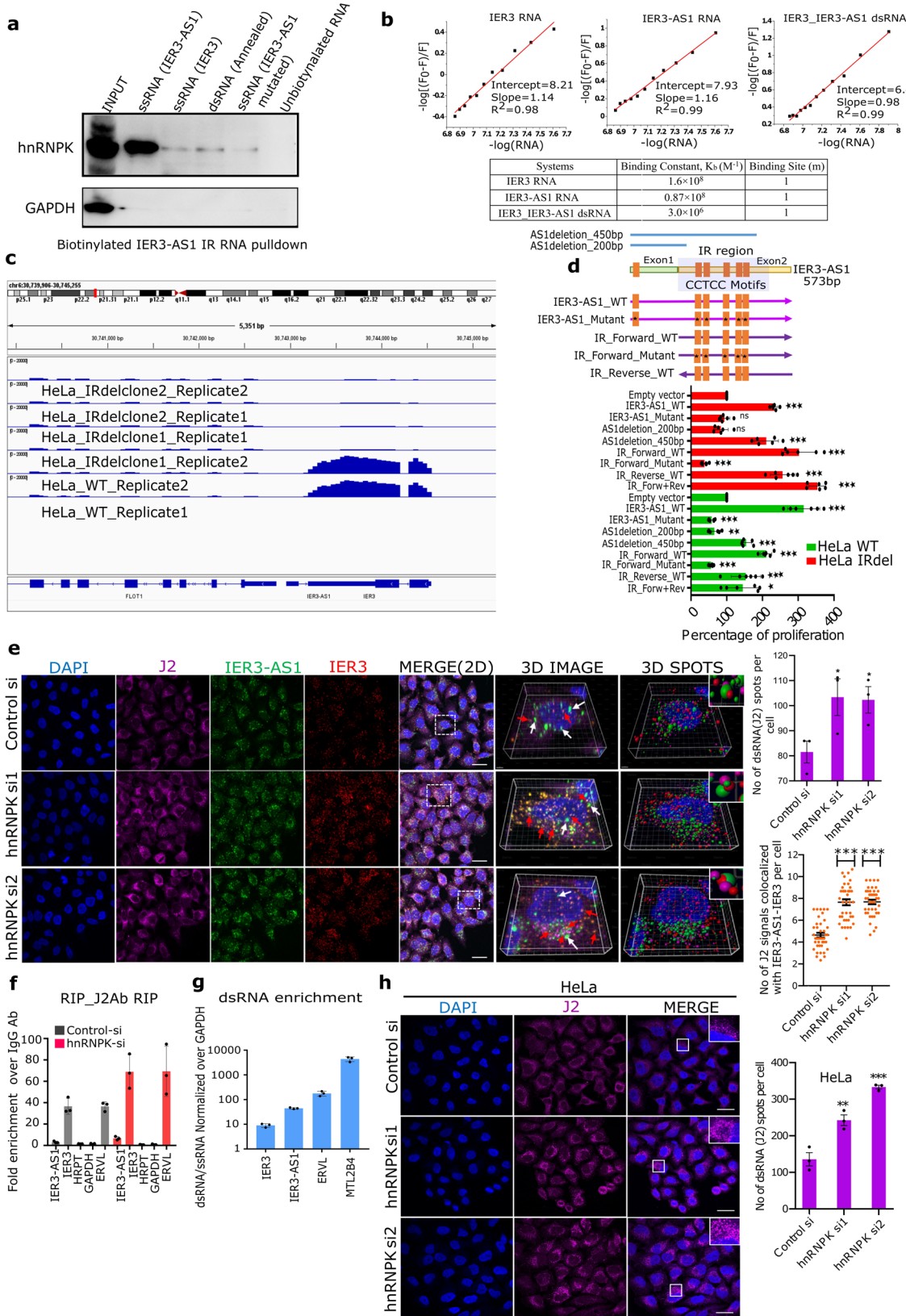

Particularly, in *hnRNPK* KD cells, increased colocalization of RNA-FISH signals of IER3 and IER3-AS1 transcripts with the dsRNA-specific immunostaining, and significant enrichment of these transcripts in the J2 antibody RIP pulldowns, clearly demonstrate the role of HnRNPK in the regulation of RNA–RNA interactions between IER3 and IER3-AS1. Increased RNA-RNA interactions between IER3 and IER3-AS1 correlate

with higher colocalization with ribosomes and an increase in IER3 translation, indicating that RNA–RNA interactions between these transcripts potentiates IER3 translation. Based on the HnRNPK tryptophan fluorescence spectra experiments with single-stranded and duplex forms of IER3 and IER3-AS1 and in vitro biotinylated RNA pulldowns followed by HnRNPK immunoblotting, it is evident that

**Fig. 5 | HnRNPK control global single- and double-strand RNA. a** Biotinylated RNA pulldowns using streptavidin beads followed by western blotting with HnRNPK antibody. GAPDH was used as a control for RNA pulldown specificity. Same results were obtained from independent replicate. **b** Plots of -log[(F0-F)/F] vs -log[RNA] for the equilibria between IER3, IER3-AS1 and duplex RNA IER3/IER3-AS1 showing the Intercept values of the HNRNPK protein ($\lambda_{ex}$ = 280 nm) with increasing amount of RNA (0.0 to 0.13 μM). Table shows the calculated equilibrium parameters. **c** BAMscale peaks over IER3/IER3-AS1 locus from RNA-seq data of control and CRISPR/cas9 IR deleted HeLa cells. **d** Schematic view of IER3-AS1 deletions and CCTCC motif mutations. Bar graph of MTT assay showing the percentage of cell proliferation. **e** IF-RNAscope images of control and *hnRNPK* KD HeLa cells showing J2 (dsRNA-specific antibody) immunostaining, coupled with RNAscope using IER3 (red) and IER3-AS1 (green) probes. The right-side panels show 3D reconstructed cell images (Imaris) of the merged image indicated by white dotted box. IER3-AS1 (green dots marked with white arrows) and J2/IER3/IER3-AS1 colocalized signals (brownish-yellow dots marked with red arrows). The spheres from the 3D reconstructed images detected using spot detection tool. J2 - Pink spheres, IER3-AS1 - green spheres and IER3 - red spheres. Upper right corner shows magnified co-localized spots in an inset. Indicative

scale bar is 50 um. Graph to the right shows the numbers of J2 signals in control (n = 81) and *hnRNPK* KD (siRNA1: n = 74 and siRNA2: n = 75) cells. Data represents mean with ±SEM from n = 3 independent experiments. *P* value calculated by One-way ANOVA (Sidak's multiple comparisons test). Dot plots shows the average number of J2/IER3-AS1/IER3 co-localized signals (n = 120 cells were manually counted from 3 independent experiments). *P* value was calculated by one-way ANOVA (Kruskal–Wallis test). **f** RIP assay, performed using J2 antibody, showing the fold enrichment of IER3 and IER3-AS1 transcripts over IgG. HPRT and GAPDH (negative controls) and ERVL (positive control). **g** RT-qPCR showing the enrichment of IER3:IER3-AS1 dsRNA (RNaseA treated) over ssRNA (RNaseA untreated). The expression values were normalized with GAPDH. ERVL and MTL2B4 are positive controls. **h** IF images of HeLa cells showing J2 signals. Magnified J2 signals (puncta) were shown in an inset indicated by white box. Indicative scale bar is 50 um. Graph shows the number of J2 signals quantified using Imaris spot detection tools. A total, control: n = 63, siRNA1: n = 51 and siRNA2: n = 57 cells were counted from three experiments. *P* value calculated by One-way ANOVA. Data represent mean ± SD from two independent biological replicates. *P* value was calculated by two-sided Student's *t*-test for (**d**, **f** and **g**) (*$p \leq 0.05$; **$p \leq 0.005$; ***$p \leq 0.0005$, ****$p \leq 0.0001$).

there is a preferential interaction of HnRNPK with single-stranded IER3-AS1 and IER3 compared to double-stranded IER3/IER3-AS1. This is indeed the case as the number of single-stranded IER3 and IER3-AS1 signals were significantly reduced upon *hnRNPK* depletion whereas there was a significant increase in single-stranded IER3 and IER3-AS1 signals following *hnRNPK* overexpression. These observations taken together highlight the function of HnRNPK in controlling the functions of overlapping sense and antisense transcripts by maintaining them in single-stranded form.

What is particularly interesting from our data is the functional connection between HnRNPK and global dsRNA formation. *hnRNPK* KD increases global dsRNA levels, whereas overexpression resulted in the loss of dsRNA. The increased appearance of dsRNA signals, as well as co-localization of IER3 and IER3-AS1 transcripts in multiple cancer cell lines following *hnRNPK* KD, point towards a critical role of HnRNPK in regulating global RNA–RNA interactions. The resulted dsRNA in the *hnRNPK* KD cells was accumulated in the cytoplasm. This was also evident in the context of IER3 and IER3-AS1, indicating that HnRNPK binding is crucial for the nuclear enrichment of IER3 and IER3-AS1 transcripts. The nuclear enrichment of RNAs has been shown to be mediated by coordinated interaction of KH domains in HnRNPK with the CCTCC motifs[16,26]. Both IER3 and IER3-AS1 transcripts harbor several of these CCTCC rich motifs in the overlapping IR region and these motifs could contribute to the nuclear enrichment through their interactions with HnRNPK. Our data clearly shows that CCTCC motifs in the overlapping region are in specific interaction with HnRNPK as IER3-AS1 lacking CCTCC motifs or harboring mutant CCTCC motifs could not rescue the defective cell proliferative and cell cycle phenotypes of the IR deleted HeLa cells. Thus, based on our data we propose that HnRNPK through its specific interactions with CCTCC motifs may maintain RNA in single stranded form and thereby precluding dsRNA formation.

Our data demonstrate that the oncogenic properties of IER3 and IER3-AS1 are determined by their interactions with HnRNPK. Both IER3 and IER3-AS1 display oncogenic properties in HEK293, HeLa and BT-549 cells. However, data on IER3 from multiple cancers indicate that it functions as an oncogene or tumor suppressor in a cancer dependent manner (Supplementary Fig. 2b)[27]. Moreover, our preliminary data from childhood cancer neuroblastoma revealed that IER3 is a p53 dependent tumor suppressor and its higher expression predicts better prognosis in multiple neuroblastoma patient cohorts. Consistent with the latter observations, the increased expression of IER3 in the *hnRNPK* KD cells did not promote oncogenic features and there was rather a significant decrease in cell proliferation and increase in cell death, indicating that hnRNPK interaction with IER3 and IER3-AS1 may determine the oncogenic functions. This notion is further supported

by the fact that hnRNPK has been shown to possess oncogenic properties in many cancers[28,29]. These observations, taken together, raise an interesting question of whether HnRNPK interaction with IER3-AS1 and/or IER3 determines oncogenic functions? Since their expression is dependent on each other, it would be difficult to discern whether IER3 or IER3-AS1 interaction with HnRNPK is crucial for their oncogenic functions. Despite an increase in the translation of IER3 in the *hnRNPK* KD cells we did not find any increase in cell proliferation of the *hnRNPK* KD cells (Fig. 5j), suggesting that IER3-AS1 interaction with HnRNPK is crucial for its oncogenic functions. Furthermore, HnRNPK showed preferential interaction with IER3-AS1 compared to IER3 and IER3:IER3-AS1 dsRNA. These observations indicate that HnRNPK determines oncogenic properties of IER3-AS1 and the loss of *hnRNPK* neutralizes IER3-AS1 oncogenic properties through the formation of dsRNA with IER3 (Fig. 7f).

## Methods

### Cell line maintenance and treatment

HEK293T, HeLa, A549 and BT-549 cells were obtained from CLS Cell Lines Service. All the cell lines were cultured using DMEM supplemented with 10% FBS and 1X penicillin and streptomycin. Human fibroblast growth factor-2 (FGF-2) (50 gn/ml) was treated for 6 h duration in duplicates for each cell line. For RNA sequencing total RNA was extracted from FGF-2 treated and control (DMSO treated) cell lines.

### Mouse xenograft tumor model

We performed all animal experiments according to the ethical permit (No: 5.8.18-02708/2017), reviewed and approved by the Animal Ethical Review Board, University of Gothenburg, Sweden. $10 \times 10^6$ stable control (Ctrl)Sh and IER3-AS1sh RNA knockdown Hela cells were subcutaneously injected on the dorsal back region of 5 to 6 week old NSG™ mice (Charles River, France) (n = 5) with 15% Matrigel (Corning). Following 2–3 weeks post-engraftment, we measured the tumor volumes. Total RNA was extracted from the tumor tissues for checking the expression of IER3-AS1, IER3 and chemokines. Formalin-fixed mice xenografts were embedded in paraffin and immunohistochemical analyses were performed on sections of 2–4 μm using an antibody against Ki-67 (proliferation marker). The dilution and reference for validation of Ki67 antibody was mentioned in Supplementary Table 5 and 6, respectively. Ki-67 images were taken using EVOS FL Auto Imaging System (Thermo Fischer scientific).

### RNA isolation, cellular fractionation and RT-qPCR

Total RNA was isolated from cell lines using ReliaPrep RNA cell mini-prep system (Promega). Nuclear and cytoplasmic fractionations of

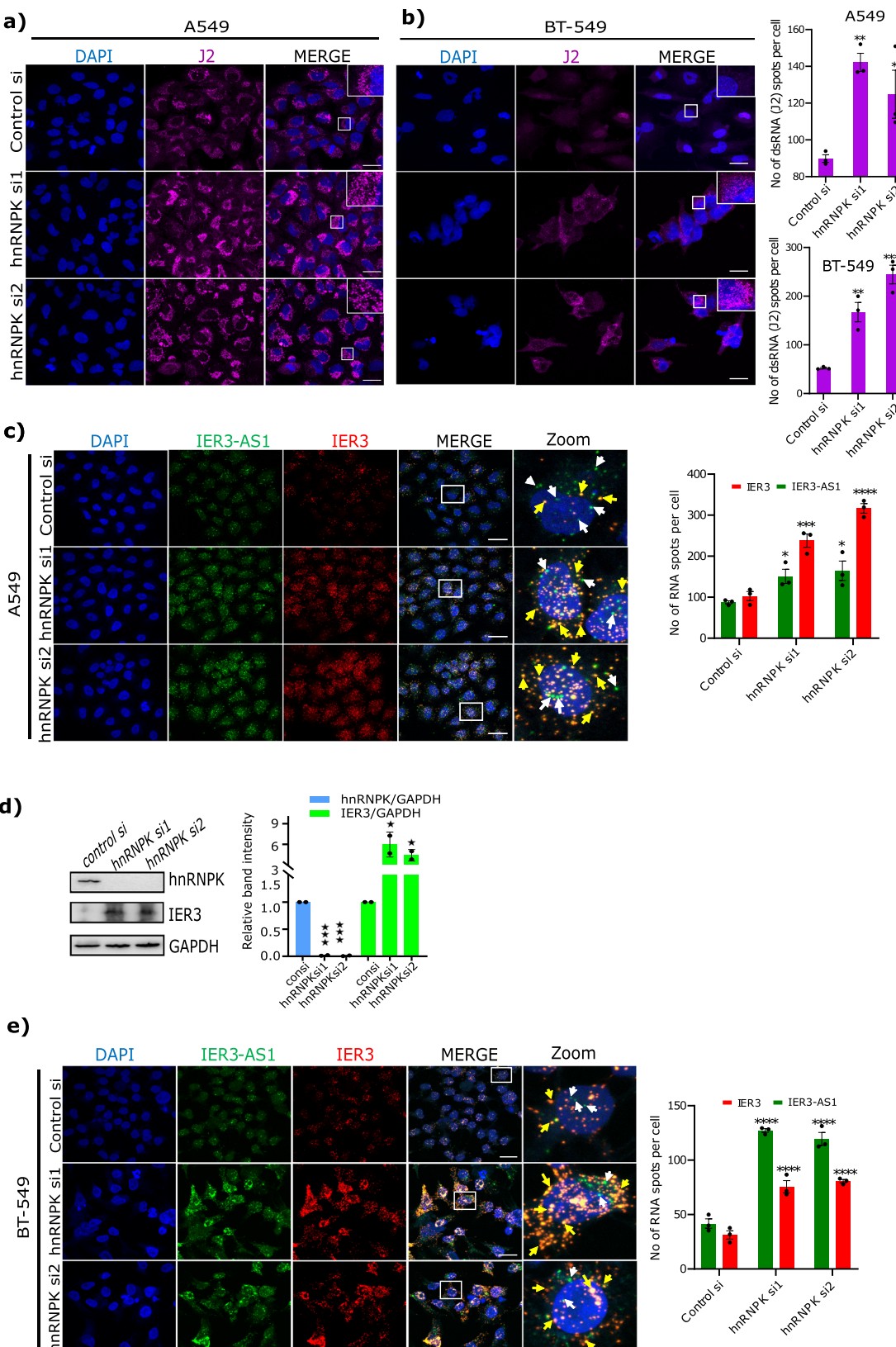

RNAs were performed as briefly described below. Cells were gently trypsinized (HeLa) and collected by centrifugation. Cell pellets were rinsed with 1X PBS and then the plasma membranes were lysed by resuspension in ice-cold Cytoplasmic lysis buffer (0.15% (vol/vol) NP-40; 5 mM Tris-HCl (pH 7.0); 150 mM NaCl; 1.5 mM MgCl$_2$; 10 U SUPERase.IN; 1x Protease inhibitor mix) for 5 min on ice and centrifuge for 2 mins at 300 g at 4 °C. The supernatant contains cytoplasmic fraction, and the nuclei pellet contains nuclear fractions. Both cytoplasmic and nuclear fractions were phenol/chloroform extracted and ethanol precipitated. Chromatin-associated RNA was purified according to the TRIzol protocol, but an additional phenol/chloroform extraction step was performed prior to precipitation. All RNA fractions

**Fig. 6 | HnRNPK induces global dsRNA formation in multiple cell lines.**
**a, b** Immunofluorescence images of A549 cells (**a**) and BT-549 (**b**) showing dsRNA-specific J2 signals in *hnRNPK* KD cells. The panel to the right shows the merged images. Magnified J2 signals (puncta) are shown in an inset at the upper right corner indicated by white box. Indicative scale bar is 50um. Graphs show the number of J2 signals in *hnRNPK* KD cells quantified using Imaris spot detection tools. For A549 cell line - control: $n = 59$, siRNA1: $n = 64$ and siRNA2: $n = 73$ cells and for BT-549: control: $n = 25$, siRNA1: $n = 37$ and siRNA2: $n = 44$ cells were counted from three independent experiments. Data are presented as mean values +/−SEM. Significance was calculated by One-way ANOVA (ordinary multiple comparisons test).
**c, e** RNAscope images of A549 cells (**c**) and BT-549 cells (**e**) showing IER3 (red) and IER3-AS1 (green) in control and *hnRNPK* KD cells. Panel to the right shows the magnified images of the merged panels indicated by white box. White arrows marks IER3-AS1 (green dots) and yellow arrows marks colocalized IER3/IER3-AS1 dsRNA (Yellow dots). Indicative scale bar on the images is 50 um. Bar graph shows the number of IER3-AS1 and IER3 RNA signals per cell in control and *hnRNPK* KD cells quantified using Imaris spot detection tools. For A549 - control: $n = 101$, siRNA1: $n = 78$ and siRNA2: $n = 97$ cells and for BT-549 - control: $n = 132$, siRNA1: $n = 44$ and siRNA2: $n = F36$ cells were counted from three independent experiments. Data are presented as mean values +/−SEM. The significance was calculated by Two-way ANOVA (Sidak's multiple comparisons test). **d** Immunoblot showing the levels of HnRNPK and IER3 in *hnRNPK* KD A549 cells. GAPDH was used as an internal loading control. Similar results were observed in an independent biological replicate. Histograms showing the relative band intensities from the western blot are shown left side. The *p* values were calculated using two-sided Student's *t*-test and data are presented ± SD from two independent replicates.

were resuspended in TE (pH 7) and quantified using a Nanodrop-1000 spectrophotometer (Nanodrop Technologies) and tested for DNA contamination by real-time RT-PCR lacking reverse transcriptase. cDNA synthesis is performed using Improm-II Reverse Transcriptase kit (Promega) and RT-qPCR analysis was done using Power SYBR Green PCR master mix (Applied Biosystems). Differences in expression were calculated using the ΔΔCt method. dsRNA enrichment was analysed using RT-qPCR analysis. RNA was then digested or not with 50 mg/ml RNase A in high-salt concentration (NaCl, 0.35 M) for 30 min. Enrichment of dsRNA over ssRNA was then calculated by normalizing the delta Ct between RNase A treated and non-treated of IER3 and IER3-AS1 (dsRNA) GAPDH (ssRNA). ERVL and MTL2B4 genes were used as positive controls. Error bars represent the SD of at least three independent experiments.

## Immunofluorescence

Immunofluorescence was performed following the protocol as described previously[30]. Detection of HnRNPK and double-stranded RNA (dsRNA) was performed after 48 h of transfection using anti-HnRNPK (1:250 dilution) and J2 antibodies (1:200 dilution), respectively. Nucleus was stained with DAPI (4′,6-diamidino-2-phenylindole). Information regarding antibodies (dilutions and references for antibody validations) used in the immunofluorescence experiments is provided in Supplementary Tables 5 and 6.

## RNAScope

RNAScope in-situ hybridization (ISH) were performed using RNAscope Multiplex Fluorescent Reagent Kit v2 (Cat. No. 323100) from Advanced Cell Diagnostics (ACD). RNA probes specific to IER3-AS1 (Cat. No: NPR-0018153) and IER3 (Cat No: NPR-0018154) were designed by ACD's made-to-order probes service. Manual RNAScope ISH protocol from Advanced Cell Diagnostics (ACD, 323100-USM) was followed to perform single or double RNA staining. RNAScope manual procedure involved sample fixation, sample pre-treatment, subsequent probe hybridization, signal amplification and ISH signal detection. Briefly, adherent cells were cultured on the glass coverslips for RNAScope ISH. The cells were fixed using 10% neutral formalin solution for 30 mins followed by washes and the pretreatment with 3% hydrogen peroxide solution ($H_2O_2$) and protease III for 10 min each at room temperature. RNA specific probes were hybridized for 2 h at 40 °C in a HybEZ oven. Coverslips were washed with 1X wash buffer 2 times 2 mins each at RT following each amplification step using reagents included in the kits. ISH detection step was completed using Opal-520, Opel-570 or Opel-690 dyes (1:1500, Akoya Biosciences) in 1X TSA buffer for 30 mins each at 40 °C in a HybEZ oven.

Dual fluorescent ISH and immunofluorescence were performed using integrated co-detection workflow (MK 51–150 manual). Upon completion of pretreatment steps, coverslips were incubated with primary antibodies at 4 °C overnight. Next day, the coverslips were again fixed with 10% neutral formalin solution for 30 mins at RT followed by protease III treatment for 10 mins at RT. RNAScope ISH

procedure were followed after the protease III treatment till HRP blocker steps then coverslips were incubated with secondary fluorescence antibody to detect primary antibody for 30 mins at RT. The coverslips were mounted with prolong gold antifade reagent with DAPI (Molecular probes, P36931). RNA in-situ's (RNAScope) and immunofluorescence imaging were performed on Zeiss LSM 700 inverted, Zeiss LSM 780 and Zeiss LSM 880 airy scan confocal microscope at the Centre for cellular imaging facility. Majority of the images were captured on 40X or 60X oil immersion objectives with the laser emitting 405, 488, 561 and 670 nM wavelength depending on fluorophores used in the experiments.

## Image analysis

**3D reconstruction.** The 3D reconstruction was performed using Imaris microscopy image analysis software 9.8.2 version. The confocal images were captured at the 0.40 uM z-step size. Z stacked raw image then converted into the Imaris compatible file using imaris file converter tool. The 3D reconstructed images were visualised at surpass mode in 3D view window. Most of the image analysis were performed on the 3D reconstructed image.

**Quantification.** Spot detection: Spot detection tool was used to visualize and measure intracellular fluorescence signals in the form of puncta or dots or particles within a cell. Spot detection was performed to quantify the number of RNA signals and their fluorescence intensity. Spot detections were performed on the 3D reconstructed images using spot creation wizard in imaris 9.8.2 version software. The detailed steps are given in the Supplementary Fig 6. Briefly, first we measured the average diameter of the puncta signal using distance measurement tool in slice function. The average diameter value was used to detect the possible spots and create the sphere of same diameter within the cell. The non-specific spots were removed by adjusting the detection ability through quality control wizard. The spot diameter value and quality adjustment values were kept constant across the samples and replicates. Finally, after the completion of spot creation wizard, the number of spots and other statistics of each spot including intensity, distance, area, volume, position etc. were quantified and statistically analysed in the graph pad prism 8.4. AS1, IER3 RNA quantification: Spot detection was performed to quantify the number of AS1 and IER3 RNA signals in the cell. We measured 0.5 um average diameter of AS1 and IER3 puncta signals in HeLa cells and used to detect the spots and create a sphere. Similarly, we used 0.5 nm for A549 and 1 nm diameter for BT549 cells to detect AS1 and IER3 signals. The spots quality above 33 for AS1 and above 16.5 for IER3 were used to remove non-specific spots. The number of spots and intensity of each spot were quantified and statistically analyzed in the graph pad prism. J2 (dsRNA antibody) signal quantification: J2 signals/puncta were quantified as a spot. Spot detection was performed to quantify the number of J2 signals in the cell. We measured 0.4 um average diameter of J2 puncta signals in HeLa cells and used to detect the spots and create a sphere. Similarly, we used

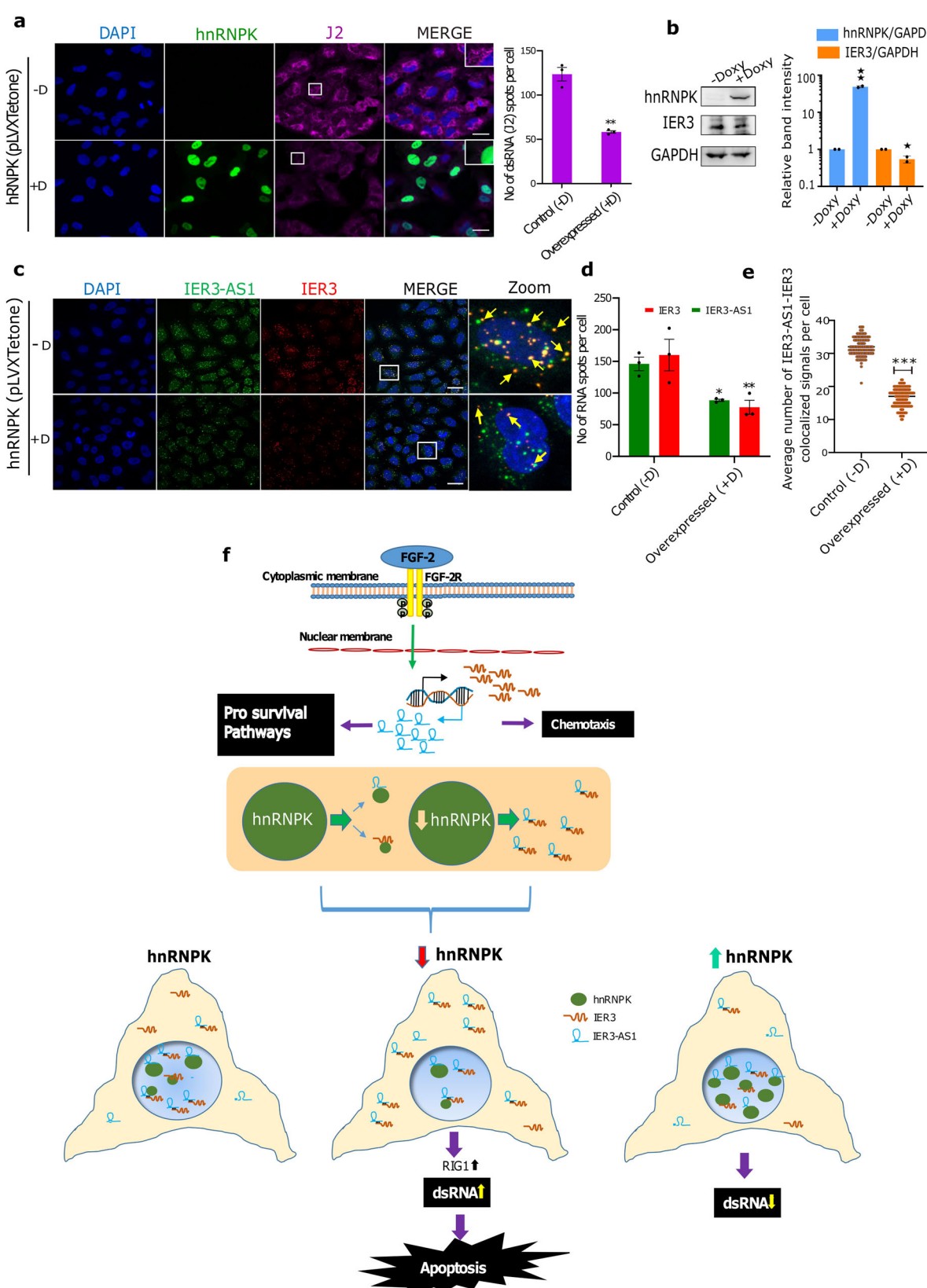

0.75 nm for A549 and 1 nm diameter for BT549 cells to detect J2 spots. The spots quality above 150 for Hela and above 320 for A549 and BT549 cells were used to remove non-specific J2 spots. The number of spots and intensity of each spot were quantified and statistically analyzed in the graph pad prism.

**Co-localization quantification.** The overlapping/colocalized signals of AS1-IER3 (dsRNA), J2-AS1-IER3 and RPS3-AS1-IER3 were quantified manually by visualising each channel signal. The number of colocalized signals were quantified and statistically analyzed in the graph pad prism.

**Fig. 7 | HnRNPK overexpression suppresses global dsRNA levels.**
**a** Representative immunofluorescence images showing the effect of doxycycline (DOX) induced overexpression (−DOX/+DOX) of HnRNPK (green) on global dsRNA as detected using J2 antibody (red) in A549 cells. Panel to the right shows the merged images. Magnified J2 signals (puncta) are shown in an inset at the upper right corner indicated by white box. Indicative scale bar on the images is 50 um. Bar graph to the right shows the number of J2 signals in HnRNPK non-induced (−DOX (−D); *n* = 81 cells) and HnRNPK induced (DOX+(+D); *n* = 53 cells) cells from three replicates quantified as RNA spots per cell using Imaris spot detection tools. Data are presented as mean values +/−SEM. Statistical significance was calculated by Two-tailed unpaired *t*-test. **b** Immunoblot showing the levels of HnRNPK and IER3 in −DOX (−D) and +DOX (+D) A549 cells. GAPDH was used as an internal loading control. The similar results were observed in an independent experiment. Histograms showing the relative band intensities of the western blot are shown left side. The *p* values were calculated using two-sided Student's *t*-test and data are presented ± SD from two independent replicates. **c** RNAscope images of DOX (−D/+D)

induced HnRNPK cells showing IER3 (red) and IER3-AS1 (green) in control (−D) and *hnRNPK* overexpressed (+D) cells. Panel to the right showing magnified images of the merged panel indicated by white box. Yellow arrows depict colocalized IER3/IER3-AS1 dsRNA (yellow dots). Indicative scale bar on the images is 50 um. **d** Bar graph showing the number of IER3-AS1 and IER3 RNA signals per cell in control (−D; *n* = 80 cells) and *hnRNPK* overexpressed (+D; *n* = 121 cells) cells quantified using Imaris spot detection tools from three replicates. Data are presented as mean values +/−SEM. Statistical significance was calculated by Two-way ANOVA (Sidak's multiple comparisons test. **e** Dot plots showing the average number of colocalized signals of IER3-AS1/IER3 (dsRNA) per cell in control (−D) and *hnRNPK* over-expressed (+D) cells. A total of 150 cells were manually counted from three independent replicates. Data are presented as mean values + / SEM. Statistical significance was calculated by Two-tailed unpaired *t*-test. For figures (**a**, **d** and **e**) **p* ≤ 0.05; ***p* ≤ 0.005; ****p* ≤ 0.0005. **f** Model explaining the role of HnRNPK in the regulation of global dsRNA.

## Immunoblot and RNA dot blot assay

For protein extraction, cells were washed in 1X cold PBS and lysed in RIPA lysis buffer (Sigma Aldrich; 20–188) and complete protease inhibitor cocktail. The cell lysates were spun down at maximum rpm for 20 min at 4 °C and the supernatants estimated for protein concentration using Pierce BCA Protein Assay Kit (Thermo Scientific; 23225) according to the manufacturer's instructions. 30 μg of total cell lysates were resolved by SDS PAGE on NuPAGE Novex 4–12% Bis-Tris Protein Gels (Invitrogen). Following the electrophoresis, proteins were transferred on to 0.45 μm nitrocellulose membrane (Hybond ECL, Amersham, GE healthcare). Membranes were blocked in 5% BSA in 1X TBS-T (10 mM Tris-base pH 7.5, 150 mM NaCl, and 0.1% Tween) for 1 h at room temperature followed by overnight incubation with primary antibody prepared in fresh blocking solution. Membranes were washed thrice for 5 min each in 1X TBS-T followed by incubation with anti-mouse or rabbit secondary antibodies for 1 h at room temperature in 5% skimmed milk made in 1X TBS-T. Membrane washes were repeated 3 times in 1XTBS-T as above followed by visualizing protein bands with chemiluminescent Substrate (Thermo Scientific) using IMAGE BioRad 4.0 alpha software.

For RNA dot blots, serial dilutions of total RNA were made using nuclease free water. Incubate the serially diluted RNA at 95 °C for 3 min and chill the tubes on ice. Dropped 2 μL of RNA samples onto the positively charged nylon membrane. Air dried and crosslinked the RNA to the membrane with UV light: 125 mJoule/cm2 at 254 nM in the chamber of SG Linker. Washed the membrane in 1X TBST (1X TBS, 0.1% Tween-20), for 5 min at room temperature with gentle shaking to wash off the unbound RNA. Then proceeded with immunoblotting with primary and secondary antibody as described above. Details of the antibodies used for immunoblotting are provided in Supplementary Tables 5 and 6.

## Plasmid constructs and molecular cloning

IER3-AS1 interacting region (IR) which is 325 bp sequence, mutant IR sequence along with full length wild type IER3-AS1 and mutant IER3-AS1 sequences were amplified from the HeLa cells using the primers described in Supplementary Table 2 and cloned into the pcDNA3.1 vector downstream to T7 polymerase site in both orientation for in vitro transcription of IER3-AS1 and IER3 IR transcripts. The CCTC motifs are mutated to AAGA sites by Genescript Bitech Ltd. For transductions, IER3-AS1and IER3 shRNAs for KD and sgRNAs for over expression were cloned in lentiviral vectors as described in Supplementary Table 3. All the clones were sequence verified before use in the experiments. Primers used in cloning are provided in Supplementary Table 2. The sgRNA sequences used for HeLa_IR CRISPR deletion and the primer sequences used for screening IR deletion clones are listed in Supplementary Tables 3 and 2, respectively.

## Transient and stable transfections

siRNAs were obtained from Sigma Aldrich and transfected using Lipofectamine RNAiMAX reagent using manufacturer instructions. The control siRNA, siRNA sequences/siRNA ids targeting different genes are described in Supplementary Table 3. Transfection of the plasmid DNA was performed using Lipofectamine 2000 reagents using manufacturer instructions. PCDNA3.1 vector is used as control for DNA transfections. Stable cell lines with KD and OE cells were performed using Lentiviral shRNA particles targeting IER3-AS1, IER3 and non-target shRNA control which were procured from Sigma Aldrich. Stable shRNA cells were generated using the method as described previously[30]. The information on sequences of the shRNAs and the vectors used for cloning is provided in the Supplementary Table 3 and 4, respectively.

HnRNPK was purchased from Addgene (Plasmid #71662) and cloned via EcoRi/AgeI restrictions sites into inducible vector pLVX-Tetone-GFP. Lenti-X Tet-One Inducible Expression System (pLVX-Tetone) were purchased from Takara Bio (Cat. No. 631847) and GFP were cloned into the empty vector to get pLVX- Tetone-GFP. A549 cells were selected in Puromycin (3 ug/ml) containing medium after transduction of pLVX-Tetone-HnRNPK-GFP. HnRNPK expression were induced by incubating cells with medium containing Doxycyclin (100 ng/ml) for 24 h. All the clones were sequenced before transductions and the down regulation and overexpression of the target genes was verified using RT-qPCR. All vectors, drugs and reagents used are listed in Supplementary Table 3 and 4.

## Cell migration/invasion and wound healing assays

The migration and invasion assay was performed using the BioCoat Matrigel Invasion Chamber (Thermofischer scientific, 11553570) as per the manufacturer's instructions. In brief, cells were seeded approximately 50,000–60,000 cells per ml in density on the upper chamber of the biocoat invasion chamber in Dulbecco's modified Eagle's starvation medium without serum. Medium with 10% serum was added to the lower chamber. After 48 h, the migrated cells were fixed and stained with DAP1 (4',6-diamidino-2-phenylindole) and images were taken in EVOS FL Auto Imaging System (Thermo Fischer scientific) and colonies were counted using imageJ software. Wound healing assay was performed using a previously described protocol by Chun-Chi Liang et al.[31].

## RNA immuno precipitation (RIP)

RIP was performed with UV crosslinked cells using the protocol described previously[32]. Briefly, HeLa cells with or without transfections were trypsinised, washed with PBS and crosslinked with 1% formaldehyde for 10 min followed by quenching with 125 mM glycine. Cells were subsequently washed twice with 1X PBS followed by cross-linking on ice with UV. Nuclei were isolated using 1X nuclei isolation

buffer (40 mM Tris-HCl pH7.5, 20 mM MgCl$_2$, 4% tritonX-100, 1.28 M sucrose), washed and resuspended in lysis buffer (0.1%SDS, 0.5% TritonX 100, 20 mM Tris-HCl pH 7.5, 150 mM NaCl, 1 ml lysis buffer per 10 million of cells) supplemented with RNasin (Promega) and subjected to sonication (Bioruptor, 20–30 cycles) to obtain chromatin fragments of about 1 kb. 50–60 μg of soluble chromatin was used in each chromatin immunoprecipitation and incubated with 5 μg of antibody. Antibody bound chromatin was washed according to our earlier published protocol with buffers were supplemented with RNasin. Protein A magnetic beads bound to immunoprecipitated chromatin were resuspended in 10X volume of elution buffer (100 mM NaCl, 10 mM Tris 7.5, 1 mM EDTA, 0.5% SDS) containing proteinase K. Proteinase K treatment was carried out at 55 °C for 45 min followed by heating at 95 °C for 10 mins to reverse crosslinking. Chromatin bound RNA was extracted with Trizol (Life Technologies) and subjected to DNase I (Promega) treatment to remove traces of DNA. The enrichment of the interacting RNAs were assayed by RT-qPCR. All primers are listed in Supplementary Tables 1 and 2 and details of antibodies used are provided in Supplementary Tables 5 and 6.

### Chromatin oligo-affinity precipitation (ChOP) - mass spectrometry (ChOP-MS)
ChOP-MS using HeLa cells was performed following the protocol as described in our earlier study[32]. In brief, around 30 million HeLa cells were used for each pull-down reaction. Cells were trypsinised, washed with 1X PBS and resuspended in 10 ml of fresh ice cold 1X PBS in 10 cm dish. The UV bulbs were pre-warmed with 200 mJ energy just before crosslinking and the petri dishes with cells were placed on ice into the cross linker and exposed to 250 mJ. Later the cells were collected in 15 ml tubes and washed once with 1X PBS, centrifuged and then resupended in 1 ml swelling Buffer (0.1 M Tris pH 7.0, 10 mM KOAc, 15 mM MgOAc, 1% NP-40, 1 mM DTT, 1 mM PMSF, complete protease inhibitor and 0.1 U/μLSuperase-in [Ambion]) in a 1.5 ml tubes, and incubated on ice 10 min. Centrifuged the cells, removed the supernatant and added 200ul of TURBO DNAse enzyme in 1X and Incubated for 30 min at 37 °C followed by 850 ul of STOP lysis solution and incubated for 15 min on ice. The chromatin was passed through the needle to homogenate nuclei properly. Nuclei pellet can be flash frozen and stored at −80 °C or further processed for chromatin preparation. Chromatin was prepared by lysing the nuclei in 0.5 ml of nuclear lysis buffer (50 mM Tris pH 7, 10 mM EDTA, 1% SDS, 1 mM DTT, 1 mM PMSF, complete protease inhibitor and 0.1 U/μl RNasin Plus RNase Inhibitor [Promega]) on ice for 10 min followed by sonication. Cell debris was cleared by centrifugation at 13,000 rpm for 15 min at 4 °C. The supernatant was then added to an equal volume of hybridization buffer (50 mM Tris pH 7, 50 mM EDTA, 1% Igepal Nonidet P40 (NP40), 1 mM DTT, 0.6 M LiCl, 1 mM PMSF, complete protease inhibitor and 0.1 U/μl RNasin Plus RNase Inhibitor [Promega]) along with the LacZ or IER3-AS1 DNA oligos (100 pmol) and hybridized for 4 h at room temperature by gentle rotation. Total 8 probes were used for each pull-down (8 probes for LacZ and 8 probes for IER3-AS1) and the sequences for all these probes were listed in Supplementary Table 3. Meanwhile 100 μl of Streptavidin magnetic C1 beads were washed for couple of times in nuclear lysis buffer followed by 1 h incubation at room temp in nuclear lysis buffer containing 500 ng/μl yeast total RNA. The beads were then washed thrice in nuclear lysis buffer and were added to the hybridization mixture and incubated for 1 h at room temperature by gentle rotation. Beads were then washed twice in Wash buffer I (10 mM Tris pH 7, 300 mM LiCl, 1% SDS, 0.5% NP40 and 1X complete protease inhibitor) and Wash buffer II (10 mM Tris pH 7, 600 mM LiCl, 1% SDS, 0.5% NP40 and 1X complete protease inhibitor) at room temperature followed by elution of the bound RNA-Protein complexes with elution buffer (10 mM Tris pH 7, 0.1% SDS and 1 mM EDTA) at 75 °C for 5 min on a shaker heat block for MS analysis. The eluted protein samples from both LacZ control probes and IER3-AS1 probes pulldown were sent for

Mass-spectrometry analysis at the Proteomics Core Facility at Sahlgrenska Academy, Gothenburg University. The list of primers is provided in Supplementary Table 1.

### Mass spectrometry analysis
Pulldown preparations were processed using the modified filter-aided sample preparation method[33]. In short, samples, in eluted buffer, were treated with 100 mM DTT for 30 min at 56 °C, transferred to 30 kDa Microcon Centrifugal Filter Units (Merck), washed 3 times with 8 M urea and once with Trypsin digestion buffer (25 mM triethylammonium bicarbonate (TEAB), 0.5% sodium deoxycholate (SDC)) prior to alkylation with 10 mM methyl methanethiosulfonate for 30 min. Samples were digested with Trypsin (0.3 μg) (Pierce MS-grade, Thermo Fisher Scientific) at 37 °C overnight and the digestion was extended for additional 2 h with fresh Trypsin. Peptides were collected by centrifugation and purified using HiPPR™ spin columns (Thermo Scientific). SDC was removed by acidification. Samples were desalted (Pierce peptide desalting spin columns, Thermo Fischer Scientific) and dried prior to reconstitution in 3% acetonitrile (ACN), 0.2% formic acid (FA).

Samples were analyzed on an Orbitrap Fusion Tribrid mass spectrometer interfaced with Easy-nLC1200 liquid chromatography system (Thermo Fisher Scientific). Peptides were trapped on an Acclaim Pepmap 100 C18 trap column (100 μm x 2 cm, particle size 5 μm, Thermo Fischer Scientific) and separated on an in-house packed analytical column (75 μm x 300 mm, particle size 3 μm, ReproSil-Pur C18, Dr. Maisch) using a gradient from 5% to 28% ACN in 0.1% FA over 75 min followed by an increase to 80% ACN in 0.1% FA for 5 min. The precursor ion mass spectra were acquired at a resolution of 120 K and a scan range of 375–1500 m/z. The most abundant ions with a charge of 2–7 were selected for data-dependent MS/MS analysis. The MS/MS analysis was acquired using HCD at normalized collision energy settings of 30, a "top speed" duty cycle of 1 s, isolation window 0.7 m/z. Dynamic exclusion was set to 10 ppm for 45 s. MS/MS spectra were recorded at a resolution of 30 K with maximum injection time set to 110 ms. Analytical blanks were recorded prior to each sample injection. The data were analyzed using Proteome Discoverer version 2.4 (Thermo Fisher Scientific). Database searches were performed with Mascot against the SwissProt human database. Precursor mass tolerance was set to 5 ppm and fragment mass tolerance to 30 mmu. Tryptic peptides were accepted with one missed cleavage. Methionine oxidation and cysteine alkylation were set as variable modifications. Decoy search against a reversed database was used for PSM validation with the strict FDR threshold of 1%. The identified proteins were grouped by sharing the same sequences to minimize redundancy and were filtered at 5% FDR.

### Determination of binding parameters using tryptophan emission
Recombinant Human hnRNPK protein (HNRNPK) was purchased from Abcam (ab132460). The interacting region (IR) of IER3 (reverse) and IER3-AS1 (forward) RNAs were prepared by in vitro transcription reactions using T7 polymerase and used for the protein to single and dsRNA binding experiments. All buffer reagents were purchased from Sigma Aldrich and were used without further purification. The buffer used was 50 mM Tris-HCl, 1 mM EDTA, 140 mM NaCl, 5 mM KCl and 5 mM MgCl$_2$ at pH 7.4. For binding constant determination, the concentration of HnRNPK was 0.1 μM and the RNA concentrations varied in the range 0.0–0.13 μM. Fluorescence spectra were recorded on a SPEX Fluorolog (Jobin Yvon Horiba) fluorimeter with tryptophan excitation at 280 nm. Emission was collected at a right angle with an integration time of 0.1 s and wavelength interval of 1 nm. Monochromator slits were adjusted to achieve optimal signal output, leading to an interval of 5 nm for the excitation and emission side. Emission spectra were corrected for Raman scattering from water by subtracting the corresponding emission from a sample containing only solvent.

A second-order polynomial Savitzky-Golay (five points) smoothing filter was applied to all spectra. All the experimental measurements were conducted at 25 °C. The equilibrium between protein-RNA complexes and free protein upon addition of RNA results in a relationship between fluorescence without RNA present ($F_O$) and fluorescence at a certain titration point ($F$) which corresponds to the complex formation between protein and RNA. This gives the following equation: $\log[(F_0 - F)/F] = \log K_b + m \log[RNA]$; where K and m are the binding constant and the number of binding sites, respectively, and can be calculated by plotting $-\log(F_0 - F/F)$ vs $-\log[RNA]$.

## Cell proliferation assay

We carried out the proliferation assay using CellTiter-Glo™ luminescent cell viability assay kit (Promega, USA) according to the manufacturer's instructions. In the case of transient or inducible KD, we assayed the proliferative cells 96 h post-treatment, respectively. In stable KD cells, we assayed the viability 48 h after seeding an equal number of cells. Statistical analyses were performed using 3 independent biological replicates to calculate the p-value derived from a two-sided unpaired Student's t-test.

## Cell colonization, cell cycle analysis and apoptosis assay

We assessed the clonogenicity of the tested cells using standard 6-well plates. We seeded individually suspended 5000 cells/well of each cell line and allowed them to grow for one week. The proliferating cells were washed with PBS, fixed with 100% methanol for 20 min at room temperature and then stained with 0.5% crystal violet in 25% methanol. Excess stain was washed away several times with dH2O, and the plates were left to dry. Stained colonies were photographed using a digital camera.

The Cell cycle profiles of different cell lines and percentage of Apoptotic cells were analysed using the NucleoCounter NC-3000 platform (Chemometec, Denmark). We checked the profiles of stable KD cells after 48 h post-seeding. The cells were collected, washed with PBS, and fixed in 70% ethanol at −20 °C overnight. We stained the fixed cells using DAPI solution (Chemometec, Denmark) for 10 min at 37 °C and analyzed the cell cycle profiles according to the manufacturer's instructions. The data was analyzed from Nucleocounter cell cycle analysis software and the graph was plotted with those values using GraphPad Prism7.

Apoptotic cells were measured using Annexin V Assay kit (Chemometec, Denmark) according to the manufacturer's instructions. In brief, 72 h post-transient transfected HeLa cells were harvested, washed, and stained with Hoechst 33342, Annexin V-CF488A and PI and analyzed using image analysis, where the NucleoCounter NC-3000 system automates quantification of early apoptotic cells based on Annexin V binding and PI exclusion. Histograms show the Annexin V-CF488A intensity of the cell population. Scatter plots show the Annexin V-CF488A intensity versus the intensity of Propidium iodide (PI).

## Chemotaxis assays

3D μ-Slide Chemotaxis assay (ibidi GmbH) was performed according to the protocol provided by the manufacturer, as described previously[34]. Cryopreserved Human Peripheral Mononuclear Cells (PBMCs) (3H Biomedical AB, Uppsala, Sweden) isolated from single healthy donor were cultured in RPMI medium supplemented with 10% heat inactivated FBS for overnight and checked for cell viability and then mixed with neutralized solution of Corning Collagen I, Rat (ThermoFischer scientific). To yield a final concentration of $10^6$ cells/ml and 1.5 mg/ml collagen, 90 μl collagen I (5 mg/ml) was mixed with 20 μl 10 x DMEM, 5 μl 1 M NaOH, 81 μl deionized H2O, 4 μl 7.5% NaHCO3 and 50 μl 1 x DMEM, and 50 μl of 6x concentrated cell suspension was added to the collagen solution. Cell-containing solution was loaded into the 3D μ-Slide Chemotaxis side in the area of observation between the chambers and the slide was placed in the incubator for cell adhesion or collagen gelation for around one hour. The reservoirs of the chemotaxis chamber on both the sides were filled with DMEM medium containing FGF-2 treated HeLa controlsh cells and Hela IER3-AS1sh KD cell suspensions at the concentration of $1.5 \times 10^6$ cells per ml. For the control slide, the reservoirs are filled with DMEM medium containing FGF-2 treated HeLa controlsh cells on both the sides. The slides were again placed in the incubator for the cells to sediment and attach for 30 min. Then the slides were placed in the Phase contrast microscope at 4X magnification. A phase-contrast video of cell migration at a 4x magnification was recorded for 6 h, with 10 min time-lapse interval on an LSM880 Airyscan Microscope from Center for Cellular Imaging, core facility, Gothenburg University. Trajectories of 35–45 cells in the observation area of each image sequence were tracked manually using the ImageJ Manual Tracking plugin. The chemotactic effect was evaluated using the ibidi Chemotaxis and Migration Tool, according to the manufacturer's instructions. Several values characterizing cell migration and chemotaxis are computed from the trajectories by the software, such as forward migration indices (FMI), velocity, and directness. FMI express the directionality of migration (i.e., the efficiency of a cell to migrate in direction of a given Advanced chemotaxis assay for slow-moving cells chemotactic stimuli). Directness expresses the straightness of the cell path, irrespective to the gradient direction. The velocity was computed as the ratio of the accumulated distance of the cell path, and the time of migration. The list of software and algorithms is provided in Supplementary Table 4.

## RNA-sequencing, differential expression and pathway analysis

RNA from *FGF-2* treated/untreated and *IER3/IER3-AS1* knockdown samples of with two independent shRNAs were sequenced in duplicate sets using Illumina Sequencing Platform. The Illumina sequenced reads were aligned to hg38 (GRCh38) version of the genome using Hisat2.2.1 aligner and quantified using featureCounts (Subread-1.4.5). The differential expression of treated and untreated samples was performed using DESeq2 using R package. Adjusted p-value of 0.05 by Benjamini-Hochberg method and log2FC of 1 is considered as significant. Functional enriched biological pathways of differentially expressed protein coding genes were determined using GeneSCFv1.1-p2[35] (FDR < = 0.05). The differentially expressed genes were considered significant with adjusted p-value of 0.05 by Benjamini-Hochberg.

## Statistics and reproducibility

Statistical analyses were performed using GraphPad (Prism 8.4). Two-tailed students t-tests were used to determine the statistical significance of gene expression, Apoptosis, cell proliferation and cell cycle analysis. RT-qPCR calculations were presented as mean ± SD values. One-way ANOVA (Kruskal–Wallis test) was used to analyze the single-stand RNA. The specific tests and statistics used in the analyses were mentioned in the figure legends. P values less than 0.05 are considered statistically significant: $*P < 0.05$, $**P < 0.01$, $***P < 0.001$, $****P < 0.0001$. All experiments were performed using 2 to 3 independent replicates. No data were excluded from the analyses.

## Reporting summary

Further information on research design is available in the Nature Research Reporting Summary linked to this article.

## Data availability

The data that support this study are available from the corresponding authors upon reasonable request. GEO accession number for data generated for this paper is GSE190212. The mass spectrometry proteomics data have been deposited to the ProteomeXchange Consortium via the PRIDE partner repository with the dataset identifier PXD035704. Source data are provided with this paper.

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

## Acknowledgements

We thank Vijay Akhade and Silke Reischl for help with FGF-2 treatments and plasmid cloning, respectively. All computations were performed on resources provided by Uppsala Multidisciplinary Center for Advanced Computational Science (UPPMAX) high-performance computing (HPC), which is part of Swedish National Infrastructure for Computing (SNIC). We thank the Centre for Cellular Imaging and Proteomics Core Facility at GU, for the help. Proteomics Core Facility are grateful of Inga-Britt and Arne Lundbergs Forskningsstiftlese for the donation of the Orbitrap Fusion Tribrid MS instrument. This work was supported by grants from Swedish Cancer Research foundation [Cancerfonden: Kontrakt no. CAN2018/591]; Swedish Research Council [2017–02834]; Barncancerfonden [PR2018-0090]; Ingabritt och Arne Lundbergs forskningsstiftelse (LU2020-0017) and LUA/ ALF to C.K., and the Swedish Cancer Research Foundation (Cancerfonden: contract no. 19 0106 Pj) to M.K.

## Author contributions

M.K. and C.K. conceptualized the work, designed the experiments, and wrote the paper. M.K. performed experiments and analyzed the data.

S.M., M.S., J.K., F.W., M.W. and B.P. performed the experiments, analyzed data and edited the manuscript. S.T.K. and S.S. performed bioinformatics analysis and prepared Figures. M.M. performed xenograft experiments. M.P.Y., A.D., performed experiments.

## Funding

## Competing interests
The authors declare no competing interests.
