## [Peer Review File · Nature Communications]

REVIEWERS' COMMENTS

Reviewer #1 (Remarks to the Author):

The authors have improved the quality of their data and provide stronger evidence in support of a role of hnRNP-K in modulating dsRNA formation at loci with overlapping transcripts. This is a very interesting area of investigation and exciting model. This manuscript is suitable for publication in Nature communications.

Reviewer #2 (Remarks to the Author):

The authors have added some further information, analyses, and data to answer the reviewers' comments - while many issues have been appropriately addressed, some previously formulated questions need to be solved prior to the consideration of this manuscript, especially regarding the specificity of IRE3 and IRE3-AS1 RNA with HnRNPK in vivo. This is essential to claim HnRNPK plays a novel physiological regulatory function.

(1) Specific comment 2. "The authors must show the HnRNPK CLIP peak/binding region across the full-length IRE3 and IRE3-AS1. The current prediction-based data is not sufficient to support their rationale in reasoning HnRNPK is involved in keeping these transcripts in single stranded form."

I am still not convinced that HnRNPK specifically interact with IRE3 or IRE3-AS1 via CCTCC motifs (only five nucleotides) in vivo. HnRNPK CLIP experiment is important as such data will support their base for the whole manuscript. Of note, CLIP experiment probing of protein-long lncRNAs binding is do-able in labs nowadays.

(2) Specific comment 4. "A resulting question is whether the stem-loop structure exists and how many of them interact with HnRNPK? A SHAPE-based assay will be very helpful."

The authors only performed computational prediction of the IER3 and IER3-AS1 RNAs, which exhibits multiple stem-loop structures. However, the key question is whether this CCTCC stem-loop can be formed in the full length IER3 and IER3-AS1 RNAs. The secondary structure probing across the full length IER3-AS1 or IER3 RNAs (NOT IER3::IER3-AS1 dsRNA) should be provided in cells. This is important as such conformational information will support their working model.

(3) Specific comment 2. "Importantly, how do IRE3 and IRE3-AS1 achieve their binding specificities with HnRNPK?"

The authors have performed bioinformatic prediction and mutagenesis approaches to suggest that all "C"s in CCTCC motif are critical for HnRNPK binding. However, it still remains unclear how HnRNPK exactly interacts with those nucleotides. It is highly recommended to perform computational modeling using the available crystal structural information of HnRNPK. Otherwise, this data is misleading.

(4) Specific comment 4. "Finally, I am curious to know whether the currently predicted CCTC motif is the most effective one to mediate HnRNPK binding."

This question was formulated previously because it was unclear how IRE3 and IRE3-AS1 achieve their

specificity with HnRNPk respectively (see also above Specific comment 2). Unfortunately, the authors did not address this most important concern directly. Whether the stem-loop structured CCTCC motif exists in IRE3 and IRE3-AS1 in cells? How many of these similar stem-loop structures exist in the full length IRE3 and IRE3-AS1 according to SHAPE probing? Do they also interact with HnRNPk?

Reviewer #2 (Remarks to the Author):

The authors have added some further information, analyses, and data to answer the reviewers' comments - while many issues have been appropriately addressed, some previously formulated questions need to be solved prior to the consideration of this manuscript, especially regarding the specificity of IRE3 and IRE3-AS1 RNA with HnRNPk in vivo. This is essential to claim HnRNPk plays a novel physiological regulatory function.

(1) Specific comment 2. “The authors must show the HnRNPk CLIP peak/binding region across the full-length IRE3 and IRE3-AS1. The current prediction-based data is not sufficient to support their rationale in reasoning HnRNPk is involved in keeping these transcripts in single stranded form.”

I am still not convinced that HnRNPk specifically interact with IRE3 or IRE3-AS1 via CCTCC motifs (only five nucleotides) in vivo. HnRNPk CLIP experiment is important as such data will support their base for the whole manuscript. Of note, CLIP experiment probing of protein-long lncRNAs binding is do-able in labs nowadays.

Reply: Our data unequivocally supports the functional connection between IER3 or IER3-AS1 and HnRNPk. The interaction data was supported by ChOP mass-spec, RNA-immuno FISH and RIP and we further validated this interaction using published RIP-seq data. More importantly, mutation of CCTCC motifs interfered with the oncogenic functions of IER3-AS1, which further strengthens the functional connection between HnRNPk and CCTCC motifs. Obtaining information at nucleotide level will be informative but certainly we need to follow the nucleotide level information to RNA-structure function. One needs to find which motif is functional and whether this functional motif's structural conformation is supported by highly ordered RNA secondary structures. Thus entire work represent beyond the scope of the current investigation.

(2) Specific comment 4. “A resulting question is whether the stem-loop structure exists and how many of them interact with HnRNPk? A SHAPE-based assay will be very helpful.”

The authors only performed computational prediction of the IER3 and IER3-AS1 RNAs, which exhibits multiple stem-loop structures. However, the key question is whether this CCTCC stem-loop can be formed in the full length IER3 and IER3-AS1 RNAs. The secondary structure probing across the full length IER3-AS1 or IER3 RNAs (NOT IER3::IER3-AS1 dsRNA) should be provided in cells. This is important as such conformational information will support their working model.

Reply: This investigation has uncovered new type gene regulation, involving HnRNPk and sense IER3/antisense IER3-AS1 pair of transcripts. We have convincingly demonstrated role of HnRNPk in double-strand RNA formation between IER3 and IER3-AS1 and also HnRNPk role in global double strand RNA formation. Thus, the high-resolution structural studies in this newly found function will represent follow up investigations and beyond the scope of the current investigation.

(3) Specific comment 2. “Importantly, how do IRE3 and IRE3-AS1 achieve their binding specificities with HnRNPk?”

The authors have performed bioinformatic prediction and mutagenesis approaches to suggest

that all “C”s in CCTCC motif are critical for HnRNP binding. However, it still remains unclear how HnRNP exactly interacts with those nucleotides. It is highly recommended to perform computational modeling using the available crystal structural information of HnRNP. Otherwise, this data is misleading.

Reply: Please see our response to comments 1 and 2.

(4) Specific comment 4. “Finally, I am curious to know whether the currently predicted CCTC motif is the most effective one to mediate HnRNP binding.”

This question was formulated previously because it was unclear how IRE3 and IRE3-AS1 achieve their specificity with HnRNP respectively (see also above Specific comment 2). Unfortunately, the authors did not address this most important concern directly. Whether the stem-loop structured CCTCC motif exists in IRE3 and IRE3-AS1 in cells? How many of these similar stem-loop structures exist in the full length IRE3 and IRE3-AS1 according to SHAPE probing? Do they also interact with HnRNP?

Reply: Please see our response to comments 1 and 2.